



# Methane cycling within sea ice; results from drifting ice during late spring, north of Svalbard

Josefa Verdugo[1,2], Ellen Damm[1], and Anna Nikolopoulos[3]

[1]Alfred-Wegener-Institute, Helmholtz-Centre for Polar and Marine Research, Bremerhaven, 27570, Germany

[2]University of Bremen, Faculty 2 Biology/Chemistry, Bremen, 28359, Germany

[3]Institute of Marine Research, P.O. Box 1870 Nordnes, Bergen, 5817, Norway

*Correspondence to*: Josefa Verdugo (maria.josefa.verdugo@awi.de)

**Abstract.** Summer sea ice-cover in the Arctic Ocean has declined sharply during the last decades, leading to changes in ice structures. The shift from thicker multi-year ice to thinner first-year ice changes the methane storage transported by sea ice into

remote areas far away from the sea ice's origin. As significant amounts of methane are stored in sea ice, minimal changes in the ice structure may have a strong impact on the fate of methane when ice melts. Hence, the type of sea ice is an important indicator of modifications to methane pathways. Our study is based on the combined sample analyses of methane concentration and its isotopic composition coupled with measurements of nutrient concentrations and physical variables performed on a drifting ice floe, as well as in the traversed water in late spring 2017, north of Svalbard. We report on different storage capacities of methane within

first-year ice and rafted/ridged ice, as well as methane super-saturation in the seawater during the drifting time. We show that the ice type/structures determine the fate of methane during the early melt season and that methane released into seawater is a predominant pathway. Thereafter, the pathway of methane in seawater is subjected to oceanographic processes. We point to sea ice as a potential source of methane super-saturation in Polar Surface Water.

## 1 Introduction

Sea ice is an important component of the Arctic system playing a significant role for gas exchange between ocean and atmosphere (Parmentier et al., 2013). However, global warming has led to a sharp retreat of sea ice coverage in the Arctic Ocean during the last decades (Screen and Simmonds, 2010; Serreze and Francis, 2006). During 2019 sea ice covered 4.15 million $km^2$ in summer, representing a decrease of 33 % compared to the 1981-2010 average (Perovich et al., 2019). The negative downward trend in Arctic summer sea ice coverage has been observed for more than 30 years (Grosfeld et al., 2016). This tendency is expected to continue

over the next decades (Stroeve et al., 2012), including a cascade of possible associated effects (Meredith et al., 2019). In particular, sea ice retreat may quickly induce enhanced methane ($CH_4$) emissions into the atmosphere due to the loss of its barrier function for sea-air gas exchange (Wahlstrom and Meier, 2014). Because the Arctic holds large natural sources of this highly potent greenhouse gas, this effect has to be considered as positive feedback of global warming. Moreover, the resulting decreased temporal flux retention of methane under the ice reduces oxidation intensity to the less potent $CO_2$ (Wåhlström et al., 2016). There is also

evidence that sea ice itself is crucial for Arctic methane cycling, e.g. as vector for stored methane, transporting it to remote areas far away from its sources (Damm et al., 2018). A major sea ice formation area in the Arctic Ocean are the Siberian shelf seas (Mysak, 2001), which have been described to comprise a significant source of methane (Shakhova et al., 2010). Hence, on these shallow coastal shelf areas, large amounts of methane released from the sediment may be entrapped in sea ice during ice formation (Damm et al., 2015).

Methane uptake in sea ice is discussed to happen in different ways, either entrapped dissolved in brine or as microbubbles directly in the ice matrix (Crabeck et al., 2019; Loose and Schlosser, 2011; Zhou et al., 2014). After its formation on Siberian shelves, sea ice charged with methane is pushed by the wind away from the source area towards Fram Strait by the Transpolar Drift Stream (TDS; Damm et al., 2018; Krumpen et al., 2019). Recent trends in ice transported by the TDS, show that the structure of sea ice has undergone substantial changes during the last years, shifting from thicker multiyear ice (MYI) to thinner and more fragile first

year ice (FYI; Zamani et al., 2019; Hansen et al., 2013; Maslanik et al., 2011, 2007). Sea ice dynamics, such as rafting or ridging with two or more ice floes piling up can cause thicker ice, which is more resilient to atmospheric and oceanic forcings (Thorndike et al., 1975). Consequently, complex rafted/ridged ice structures remain impermeable longer during summer melt than the younger and simpler FYI. However, data on the variation of methane content with Arctic sea ice types are still missing.

In what follows we provide new insight in how different ice structures (rafted/ridged ice and FYI), impact the pathways of methane

in sea ice as well as seawater underneath during the Arctic winter-spring transition. The study is based on a suite of $CH_4$



measurements conducted on an ice floe along 12 days of drift. Using the fact that, during the early stage of melt, thinner FYI becomes permeable faster than thicker ice allow us to highlight physical processes involved in the $CH_4$ distribution and aging internally within the ice. We discuss the circumstances for sea ice to air emission and release into seawater as potential methane pathways which in turn infer the final methane sinks. In addition, we refer to the influence of variations in hydrographic conditions

for tracing sea ice-released methane in the underlying waters.

## 2 Material and methods

### 2.1 Ice camp

During the PS106.1 expedition in 2017, RV *Polarstern* was anchored to an ice floe and drifted with the floe for 12 days (Macke and Flores, 2018). The drift started north of Svalbard at N 82° 57.7', E 10° 14.6' on 3 June and finished at N 81° 43.8', E 10° 51.4'

on 15 June, see the ice drift trajectory in Fig. 1b. The drifting speed of the ice floe was, as calculated from the ship-GPS track, 0.09 m s$^{-1}$ on average and peaked to 0.30 m s$^{-1}$ around midnight of June 11, coincident with relatively strong winds from the west-southwest (max wind 10.4 ms, average 8.8 m s$^{-1}$, average direction 252°). The ice floe was nearly circular, measuring approximately 4.1 km x 3.7 km. Once the selected ice floe was reached, an ice camp was established for a daily sampling on the ice. In total, nine ice cores were taken at eight different locations within a radius of 1.2 km around the vessel (Fig. 1d).

### 2.2 Sea ice sampling on the floe

Ice cores were taken using a Kovacs Mark II 9 cm drill ice corer. At each sampling station the first ice core was taken for *in situ* temperature measurements by inserting a needle type temperature sensor into holes that were drilled into the ice core every 10 cm using a power drill. A second ice core was taken for nutrients, methane concentration, stable carbon isotopic signature of methane (hereafter, $\delta^{13}C$-$CH_4$), and salinity. The core was immediately brought into the onboard freezing container (T < -20° C) and cut in

10 cm pieces using an electrical saw. Every piece of ice was immediately brought into a gas tight bag avoiding contact with the atmosphere. The sea ice samples were subsequently melted onboard in a 4° C cold dark room until the ice was melted. Afterwards, 120 mL glass vials were filled up with the melt water for methane concentration and $\delta^{13}C$-$CH_4$ (samples were taken in duplicates or in some cases triplicates, depending on the melted water volume) and measured following the same procedure as for the seawater samples (see 2.3). Unfiltered nutrients samples were taken in 15 mL *Falcon* tubes at the same ice depth as methane concentration

and the $\delta^{13}C$-$CH_4$ samples, and stored at -20° C in darkness. At the home laboratory, the samples were melted and analyzed for nitrate+nitrite, phosphate, silicate, nitrite and ammonia on a four channel Seal Analytical Nutrient Auto-Analyser 3 (AA3, Grasshoff et al., 1983). Ice permeability was estimated by the brine volume fraction (BVF), calculated following Cox and Weeks (1983) for ice temperatures below -2° C and Leppäranta and Manninen (1988) for temperatures above -2° C. Layers that had a BFV above 5 % were classified as permeable ice (Golden et al., 1998). Methane concentration, $\delta^{13}C$-$CH_4$, and nutrients were

measured in bulk ice. Brine samples were collected using the "sackhole" technique (Gleitz et al., 1995; Damm et al., 2015), drilling into the ice to a depth of approximately 20 cm at C8, C9, C10 and C11.

### 2.3 Seawater sampling at the edge of the ice floe during the drift

Vertical profiles of conductivity, temperature, fluorescence, and oxygen were measured daily with a ship board Sea-Bird Scientific SBE911plus CTD (Conductivity Temperature Depth profiler) equipped with ancillary sensors and integrated with a SBE32

Carousel Water Sampler with 24 Niskin bottles of 12 L each (Macke and Flores, 2018). The CTD data were postprocessed to 1 m vertical resolution according to standard post cruise processing and calibration procedures, and with help of additional water samples drawn from the Niskin bottles for onboard salinity analysis with an Optimare Precision Salinometer (Nikolopoulos et al., 2018). For the hydrographic parameters we refer to the International Thermodynamic Equations of Seawater (TEOS-10) framework (IOC, SCOR and IAPSO, 2010) with temperature as conservative temperature CT (°C) and salinity as absolute salinity

$S_A$ (g kg$^{-1}$). Within our study area, absolute salinity values exceed practical salinity values by about 0.16 and conservative temperature exceed potential temperature by about 0.003° C. Discrete seawater samples for methane concentration and for the $\delta^{13}C$-$CH_4$ were collected at different depths throughout the water column using the CTD water sampling carousel. Bubble free water samples were taken in 120 mL glass vials using a *Tygon* tubing, impermeable for gases and sealed directly with rubber stoppers and crimped with aluminum caps. Duplicate samples for methane concentration were taken at each depth and measured

onboard a couple of hours after the sampling. A 5 mL headspace was created by addition of $N_2$ gas into the vials, and then equilibrated for 1 h at room temperature. Afterwards 1.5 mL gas sample was taken from the headspace and injected into a gas



chromatograph (Agilent GC 7890B) with a Flame Ionization Detector (FID). For gas chromatographic separation a packed column (Porapac Q 80/100 mesh) was used. The GC was operated isothermally (60° C) and the FID was held at 200° C (Damm et al., 2018). Four sets of gas mixtures (4.99, 10.00, 24.97 and 50.09 ppm) were used for calibration. The standard deviation of duplicates analyses was 5 %. The methane saturation was calculated by applying the equilibrium concentration of methane in seawater related to temperature and salinity values (using the CTD 'bottle-file' upcast data) at the corresponding sampling depth following Wiesenburg and Guinasso Jr. (1979). An atmospheric mole fraction of 1.91 ppb was used, i.e. the monthly mean from June 2017 (Data provided by NOAA Global sampling networks, sampling station Zeppelin station, Spitsbergen, http://www.esrl.noaa.gov). An additional glass bottle was taken for measuring the $\delta^{13}$C-CH$_4$ and those samples were collected following the same procedure as for methane concentration, but in this case, additionally poisoned with mercury chloride (300 µL of saturated HgCl$_2$) to stop all microbial activity. The samples were kept in a 4° C cold dark room until measured at the home lab. Consequently, 25 mL of N$_2$ was added into the vials, and then equilibrated for 1 h at room temperature. Afterwards, 20 mL of sample was taken from the headspace and injected into the PreCon coupled with a Delta XP plus Finnigan mass spectrometer. Within the PreCon the extracted gas was purged and trapped to pre-concentrate the sample. All isotopic compositions were given in δ notation relative to the Vienna Pee Dee Belemnite (VPDB) standard.

### 2.4 Water velocities

A number of instruments where deployed on/through the ice for continuous measurements throughout the drift. In this study we use water current data from two ice-tethered upward looking broadband WorkHorse Acoustic Doppler Current Profilers (ADCP; Teledyne RD Instruments), deployed about 100 m from the ship and ice edge (between C7 and C4 in Fig. 1d). Both instruments were placed on the same mooring at 11 m depth (1228.8kHz, 0.5 m cell size), and 101 m depth (307.2 kHz, 4 m cell size), respectively, recording at a 3-min interval (one ping s-1, in 50-sec ensembles). The data were post-cruise quality controlled with help of the IMOS Matlab toolbox provided by the Australian Ocean Data Network (AODN) and Integrated Marine Observing System (AODN IMOS; https://github.com/aodn/imos-toolbox). The velocities were corrected for the ice drift and thereafter smoothed with a 1h-low pass filter but otherwise not further processed before use here (hence, still holding the 12 and 24 h tidal signals which are prominent in this region, e.g. Plueddemann, 1992; Fer et al., 2015).

## 3 Results

### 3.1 Sea ice core characteristics

The ice floe was formed by First Year Ice (FYI) and ridged/rafted ice. The ice thickness at the sampled stations highly varied between 90 and 280 cm while snow thickness on top of the ice varied from 0 to 90 cm (Table 1). Of the nine ice cores sampled across the ice floe (Fig. 1d), eight were taken in the ridged/rafted site, for a better understanding about the methane cycling within those complex structures. In regards to the origin of our ice floe, the backward drift trajectories and sea ice observations of our ice floe, show that the sea ice formation area may have been in the Siberian Sea and has an age of 1 to 3 years, respectively (Wollenburg et al., 2020). Vertical profiles of temperature, salinity, BVF, NO$_3$⁻, CH$_4$ concentration and the $\delta^{13}$C-CH$_4$ for all ice cores are shown in Fig. 2 with additional information in Table 1. Following Golden et al. (1998), permeable ice was classified when BVF is above 5 % (see methods).

### 3.1.1 First Year Ice

Station C3b. In situ sea ice temperature were almost homogenous towards ice bottom (< 100 cm) and varied only between -1.8 and -1.7° C (Fig. 2). Salinity ranged from 3.7 to 5.8 with the highest values at 20 cm, homogenous from 40 cm down to the ice bottom. The BVF varied between 10 and 15 %, with the upper part of the core (0-40 cm) being more permeable than the lower part. Nitrate concentration ranged between 0.24 and 0.87 µmol L⁻¹ and it slightly increased at the bottom of the ice. Methane concentration ranged from 4.7 to 5.5 nmol L⁻¹, with higher values at 40-50 cm and at 80 cm. The $\delta^{13}$C-CH$_4$ values varied from -49.09 to -42.89 ‰ with no clear pattern.

### 3.1.2 Ridged/rafted ice

Station C1. In situ sea ice temperature ranged from -3.7 to -1.7° C following a C shaped pattern with two maxima, one near the





top and one at the bottom of the ice (Fig. 2). Salinity varied from 3.7 to 6.2 with the highest values found at the top and bottom of the ice. The BVF ranged from 5 to 15.5 %, with a permeable layer (> 5 %) at the top and at the bottom of the ice, and a nearly impermeable layer in the middle. Permeability starts to increase below 50 cm towards ice bottom. Nitrate concentration ranged from 0.2 to 1.6 µmol L$^{-1}$ and it increased with ice depth. Methane concentration ranged between 4.5 and 5.5 nmol L$^{-1}$ with values more constant at depth of impermeable layers (30-70 cm). The δ$^{13}$C-CH$_4$ values ranged between -47.24 and -41.05 with more

enriched values in $^{13}$C at the depth of the impermeable layers.

Station C4. In situ sea ice temperature varied from -2.4 to -1.7° C, with the maximum value at 110 cm (Fig. 2). Salinity values varied from 3.7 to 7.3 showing the highest values at the top and bottom of the ice. The BVF varied between 8.2 and 18.3 %, i.e. permeable throughout the ice core. Nitrate concentration varied between 0.3 to 1.77 µmol L$^{-1}$ and the highest values were observed

at 210 cm. Methane concentration ranged from 4.7 to 5.3 nmol L$^{-1}$ and remained almost constant down to the bottom, with comparable concentrations to C3b and C1. The δ$^{13}$C-CH$_4$ values fluctuated highly from -53.12 to -42.59 ‰ showing a trend of being more enriched in $^{13}$C with increasing depths.

Station C6. In situ sea ice temperature varied from -4.3 to -2° C with the lowest values in the middle 90-160 cm of the ice core

(Fig. 2). Salinity highly varied between 4.3 and 11.7 showing a general increase from top to bottom with the highest values between 100-170 cm. Over the cold middle layer, salinity was high with a pronounced peak of 11.7 at 150 cm. The BVF varied between 5.6 and 15 % with the highest permeability in the middle 90-160 cm. Nitrate concentration ranged between 0.39 to 2.5 µmol L$^{-1}$ and exhibited highly variable values towards the ice bottom. Methane concentration ranged from 2.67 to 5.6 nmol L$^{-1}$ with homogenous distribution towards ice depth, except of a spike of 5.6 nmol L$^{-1}$ at 140 cm. The δ$^{13}$C-CH$_4$ values ranged from -44.57

to -38.29 ‰ with more enriched values in $^{13}$C than in stations C3b, C1 and C4.

Station C7. In situ sea ice temperature varied from -2.6 to -1.8° C with a slight increase with depth (Fig. 2). Salinity ranged from 4.6 to 8.2 with the highest values found in the upper 40 cm and nearly homogenous below that. The BVF varied between 9.7 and 18.4 % with the highest permeability at the top and at the bottom of the ice. Nitrate concentration ranged from 0.05 to 0.3 µmol L$^{-}$

$^1$ with highest values in the upper 60 cm of the ice. Methane concentration ranged from 4.4 to 5.0 nmol L$^{-1}$ with no clear pattern. The δ$^{13}$C-CH$_4$ values ranged from -45.54 to -39.06 ‰ showing a trend of being more depleted in $^{13}$C with ice depth.

Station C8. In situ sea ice temperature varied from -2.1 to -0.2° C with the lowest values in the middle 70-120 cm of the core (Fig. 2). Salinity ranged from 0.5 to 4.3 showing a general increase with ice depth and with the highest value at 110 cm and at the bottom

of the ice. The BVF varied between 3.6 and 22 % with a peak at 50 cm. Nitrate concentration ranged between 0.02 to 1.64 µmol L$^{-1}$, with highest values between 10-20 cm and it decreased below that. Methane concentration ranged from 4.5 to 5.2 nmol L$^{-1}$ showing a homogenous distribution through the ice core, comparable to C3b and C4. The δ$^{13}$C-CH$_4$ values ranged from -47.48 to -40.65 ‰ with more enriched values in $^{13}$C at the very top and at the bottom of the ice.

Station C9. In situ sea ice temperature varied from -2.3 to -0.5° C with a peak at 30 cm, but otherwise homogenously distributed through the ice (Fig. 2). Salinity varied between 2.0 and 5.7 with the lowest value observed at 40 cm, it increased below that. The BVF varied between 7.5 and 32.6 %, with the highest value at 30 cm. Nitrate concentration ranged from 0.48 to 2.98 µmol L$^{-1}$, showing a heterogeneous profile with higher values at the top. Methane concentration ranged from 3.5 to 5.2 nmol L$^{-1}$ with a decreased up to 80 cm, it increased below that. The δ$^{13}$C-CH$_4$ values ranged from -48.04 to -42.66 ‰ with more enriched values

in $^{13}$C between 50 and 140 cm.

Station C10. In situ sea ice temperature (no measurements below 140 cm) varied from -1.7 to -0.1° C with higher values at the top of the ice (Fig. 2). Salinity varied from 1.1 and 6.2 with the lowest values coinciding with the maximum temperatures in the upper part of the ice The BVF varied between 10 and 59 %, showing the highest permeability in the upper 70 cm of the ice. Nitrate

concentration ranged between 0.04 to 1.7 µmol L$^{-1}$ showing an increased in the lower part of the core. Methane concentration ranged from 3.9 to 4.9 nmol L$^{-1}$ with a general decrease towards the ice bottom. The δ$^{13}$C-CH$_4$ values ranged from -44.75 to -37.89 ‰ with a homogenous distribution towards ice depth, except of the layer between 140 and 180 cm, where more enriched values in $^{13}$C were observed with a peak at 160 cm.





Station C11. In situ sea ice temperature varied from -1.8 to -0.8° C with higher temperatures in the upper 30 cm, but otherwise with homogenous distribution below that (Fig. 2). Salinity ranged from 0.8 to 6.5 with a large variation in the upper 90 cm, and increasing with ice depth below that until the ice bottom where a decrease was observed. The BVF varied between 3 and 21 % showing heterogeneous distribution in the upper 80 cm, below it increased towards ice depth. Nitrate concentration ranged from 0.15 to 2.51 µmol L$^{-1}$ and it increased towards the ice depth with a peak at 30 cm. Methane concentration ranged from 3.8 to 5.1

nmol L$^{-1}$, with a general increase down to 180 cm, and decreasing concentrations below that. The $\delta^{13}$C-CH$_4$ values ranged from -47.42 to -42.41 ‰ being more enriched in $^{13}$C at the bottom of the ice, where lower methane values are found.

**3.2 Hydrographic characteristics of the seawater**

During the ice drift, the bulk of waters in the upper 100 m were characterized as Polar Surface Water (PSW, $\sigma_0 < 27.70$ and $\theta < 0°$ C, see e.g. Rudels et al., 2000). The temperature (CT) was generally close to freezing temperature and varied between -1.82 and -

1.65 (average -1.79° C) down to 60 m depth while increasing steadily below that, to temperatures ranging between -1.34 and -0.13° C (average -0.75° C) at 90-100 m depth (Fig. 6a and 7a). The salinity (S$_A$) varied from 33.82 to 34.39 (average 34.31) in the upper 60 m, and between 34.45 and 34.63 (average 34.53) at 90-100 m depth. The freshest salinities where observed at stations 24-1 – 27-6 (sampled 7-10 June) in connection to slightly increased temperatures in the upper 40 m. The upper 60 meters of the water column were relatively weakly stratified, yet exhibiting alternating patches of more or less stable stratification as shown by

the Brunt-Väisälä (buoyancy) frequency (Gill, 1982) in Fig. 7b. The observed conditions were in line with earlier reported values from this area and season (Meyer et al., 2017; Rudels et al., 2000).

To help us characterize the stations which, in general, showed only the first signs of seasonal melt (and a few still with winter conditions prevailing) the mixed layer depth (MLD) was calculated for two density thresholds (Meyer et al., 2017); a difference of 0.003 relative to 3 m depth indicative of the in-season MLD formed by the first melting, and a difference of 0.01 kg m$^{-3}$ relative

to 20 m depth indicative of the depth of the past winter convection, see Fig. 7b. The melt-affected MLD was 19 m on average over the entire drift while the deeper MLD averaged to 37 m. However, the variation between stations was rather large (std dev ~ 11 m) for both these estimates. Among the stations sampled for methane, stn22-2 hosted the deepest "winter-layer" (66 m) while stn27-6 hosted the shallowest (26 m).

The salinity and temperature characteristics were used in the winter to summer transition formula of Peralta-Ferriz and Woodgate

(2015; their equation (2)) to estimate the ice thickness required to transform a winter mixed layer into a thinner fresher summer mixed layer, Fig. 7a. This calculation naturally makes better sense by the end of the melting season but was used here as a quick means to compare the ice-melt status at our stations, as found by our observed seawater properties and the values of sea ice density = 920 kg m$^{-3}$ and sea ice salinity ~ 6 given in Peralta-Ferriz and Woodgate (2015). The largest amount of melting was indicated at stations 27-6 and 24-1, coincident with the freshest salinities and with temperatures exceeding freezing temperature the most (Fig.

215 8).

**3.3 Methane concentration, saturation and the $\delta^{13}$C-CH$_4$ in the seawater**

In the seawater, the methane concentrations varied from 3.3 to 4.8 nmol L$^{-1}$ corresponding to saturations between 90 and 120 %, relative to the atmospheric background. In general, the highest methane concentration was observed during the first part of the drift, over the Yermak Plateau (YP). During the latter part of the drift over deeper waters along the slope, methane concentration

decreased. The $\delta^{13}$C-CH$_4$ values in the seawater ranged between -44.17 and -38.73 ‰ VPDB. A heterogeneously distribution was observed, with no clear pattern. Nevertheless, values more enriched in $^{13}$C coincided with higher methane saturation (Fig. 6b).

**4 Discussion**

Our study traces the methane pathways within drifting sea ice, and between the sea ice and the seawater underneath. The campaign took place from 4$^{th}$ to15$^{th}$ June 2017, over a relatively small geographic area of the Yermak Plateau, but with slightly different

characteristics between two 'regions' (Fig. 1b). Our drift started in the northeastern, relatively shallow parts (depth 800-1000 m; here denoted Region 1) of the Yermak Plateau. Windful days on 9-11 June, brought us into deeper waters over the eastern flanks of the plateau. During the last days, June 11-15, we drifted southwestward along the slope (depth 1300-1500 m; Region 2) until it was time to abandon the floe and return to harbor.



Our ice floe consisted of both thin FYI and rafted/ridged ice of various thicknesses and internal structure, and this heterogeneity among our ice cores was reflected by differences in the melting process within them. In this context, FYI was permeable throughout the entire ice column, while deeper segments of complex structures like rafted or ridged ice were still impermeable. We used ice permeability as indicator of stage of melt to follow the methane pathways, as ice permeability determines the capacity for methane storage in sea ice. Hence, impermeable layers may hold relict winter conditions, while permeable layers reflect the ongoing melt, i.e. the current early spring conditions.

We first discuss potential initial (source) methane signals still preserved within impermeable layers all since it was trapped in the sea ice. Second, we follow the pathways of methane when melt starts, focusing on the sea ice interactions with the seawater underneath the ice floe. Finally, we discuss methane saturation deviations in the upper 100 m of the water column along our drift path.

### 4.1 Fate of methane transported by different sea ice types

When the brine volume fraction (BVF) drops below approximately 5 %, sea ice becomes impermeable (Golden et al., 1998), leading to restricted gas exchange (Loose et al., 2017; Rutgers van der Loeff et al., 2014). Building off this principle, we searched for impermeable layers as relicts of winter conditions to highlight the fate of methane enclosed within drifting ice. We detected winter (i.e. pre-melt) conditions in two different types of "sandwich structures": i) An impermeable layer in the middle of the ice separated by permeable layers on the top and bottom of the ice, ii) A permeable layer enclosed by impermeable layers on the top and bottom of the ice (Fig. 3).

#### 4.1.1 Methane source signals in relicts of impermeable sea ice

In isolated impermeable layers, the methane concentration was higher relative to the atmospheric background concentration. In addition, the stable carbon isotopic signature deviated from the atmospheric background value (see C1 and C11 in Fig. 2, , and Fig. 4). As both results corroborate restricted gas exchange during the ice drift, we suggest that methane preserved in these impermeable layers has been enclosed during ice formation. Methane uptake in sea ice is reported to happen by freeze up events of super-saturated seawater (Crabeck et al., 2014; Damm et al., 2018). Thus, the initial methane inventory trapped in impermeable layers may have its source far from the present location of the ice floe. Large fractions of sea ice that reaches the Fram Strait is originated from the Laptev Sea (Krumpen et al., 2016, 2019), while the Siberian shelf waters are known to be super-saturated with methane (Thornton et al., 2016). Furthermore, the methane concentration in waters covered by sea ice in the Laptev Sea shelf area can be up to three orders of magnitude higher than background concentrations (Sapart et al., 2017). The origin of the excess methane is microbial, produced in sediments and partially oxidized before reaching the seawater (Sapart et al., 2017). We attribute the offset in stable isotope ratios in our sea ice compared to the seawater above the sediment from Sapart et al. (2017) to either fractionation that occurred during freeze up or microbial methane consumption that took place in the seawater before uptake into sea ice. To address these hypotheses, future studies should directly compare both sea ice and water, particularly during ice formation.

#### 4.1.2 Methane oxidation in permeable sea ice protected by impermeable layers

We observed in a complex ridged/rafted ice structure an enclosed permeable layer surrounded by impermeable sea ice (Fig. 2, C6, Fig. 3). This type of ice structure is described to be formed by flooding events due to storm induced floe break up and ridge formation during subsequent floe consolidation. During those events, seawater becomes trapped in rafted ice structures creating a more saline environment within certain layers therein (Provost et al., 2017). Subsequently, the enhanced salinity keeps those layers sustained permeable protected by impermeable sea ice enveloping it. Within those enclosed permeable ice layers, we detected methane enriched in $^{13}$C compared to the source methane trapped in impermeable sea ice. As, the methane concentration correspondingly dropped down, we conclude that methane consumption has occurred in this protected saline environment during the journey of the floe (Fig. 4). During microbial methane consumption, isotopic fraction occurs as methane $^{12}$C is preferentially used compared to $^{13}$C, which in turn induces a $^{13}$C enriched residual methane pool when consumption ceases (Coleman et al., 1981).

A potential isotopic fractionation during methane consumption is corroborated by a Rayleigh curves calculated as follows Eq. (1):

$$\delta^{13}C - CH_4 = 1000 * \left(\frac{1}{\alpha} - 1\right) * \ln f + (\delta^{13}C - CH_4)_0, \tag{1}$$





α is the isotopic fractionation factor, $f$ is the fraction of the residual methane remaining in the enveloped permeable layer, and the initial isotopic composition ($\delta^{13}$C-CH$_4$)$_0$ corresponds to the isotopic composition of methane detected in impermeable layers (source signal). Using a Rayleigh fractionation model, we assume that the methane reservoir has no further sinks or inputs and no mixing occurs (Mook, 1994). In summary, the pockets of permeable ice locked by impermeable ice may act as a favorable microbial environment for methane consumption. With changes in sea ice dynamics, more of this complex ice structures may be formed, which in turn may promote changes on the methane cycling within sea ice.

**4.1.3 Fate of methane when melt starts**

The vertical distribution of impermeable and permeable layers during ongoing ice melt has been shown to be associated with different methane pathways (Zhou et al., 2014). As the melting front advances vertically trough the ice, the brine networks within the sea ice expand, transporting the methane dissolved in the brine downward. Important to note is that when snow cover diminishes, and the sea ice surface is permeable, sea ice-to-air-emissions are most likely enhanced, and methane initially entrapped

in the sea ice may be released to the atmosphere. Accordingly, the sea ice permeability combined with the snow thickness on top of the ice is particularly important, as it determines the variations in methane fluxes across the sea ice/air interface (He et al., 2013). However, at the time of our sampling, we detected impermeable layers on top of the ice and covered by thick layer of snow (e.g. Table 1; Fig. 2, C11) we therefore assume that the sea ice-air flux is inhibited at this stage of the melt. In contrast, at the bottom of the sea ice, brine is released into the ocean when basal melt starts (Eicken, 2002) discharging also the methane dissolved in the

brine into the seawater (Damm et al., 2015b). Hence, increased ice permeability at the ice bottom onsets the methane release. This circumstance eventually causes a methane depletion where low methane concentration occurs at the bottom of the ice, a scenario evident at C11 (Fig. 2). Additionally, it coincides with an increase in nitrate at the same ice depth. The correlation of these variables in C11 corroborates our above assumption (Fig. 5). Remarkably, the nitrate concentration was five times higher in the permeable layers at the bottom of the ice core compared to the impermeable layers on top of the ice, which infers a source associated to

flushing of seawater into the sea ice. Flushing here refers to the "washing out" of salty brine by relatively fresh surface melt water that percolates into the pore space (Untersteiner, 1968; Vancoppenolle et al., 2013). In addition, flushing is induced by the mechanical entrainment of the water adjacent to the ice due to the rough ice bottom being 'dragged' on the ice at different speed and manner. Nitrate availability in seawater direct underneath the ice during the time of the ice drift (5 µmol L$^{-1}$, at 2 and 3 m), further supports the possibility of enhanced concentrations at the ice bottom, due to the flushing events, washing out the remaining

methane from the ice. In this context, we conclude that methane discharge into the ocean is likely to be the preferential pathway at this time of the year (Fig. 3).

Furthermore, it is also necessary to refer to the methane pathways in the opposite scenario, i.e. when ice gets fully permeable. We stated above that the impermeability combined with the snow coverage may determine the final fate of methane, and in that context, we detected that methane as well as nitrate concentration is homogenously distributed within the ice (e.g. C4, Fig. 5). This

circumstance is also triggered by flushing events, but unlike C11, super-saturated seawater is here flushed into permeable ice and consequently, concentration of both is enhanced therein. The inexistent correlation between the variables corroborates this assumption (Fig. 5).

However, under the 'extreme' scenario of highly permeable ice, i.e. latest stage of melt, no snow coverage on top of the ice, and still methane therein, the sea ice-to-air-emissions would still need to be considered.

**4.2 Dissolved methane in Polar surface water (PSW)**

Methane dissolved in the upper 100 m of the water column was not equilibrated related to the atmospheric background values. Indeed, methane was found across the range from slightly under- to super-saturated related to the saturation capacity of seawater at the Yermak Plateau (YP, 100 %, Fig. 6b), calculated with in-situ T= -1.2° C and S_P= 34.19, which corresponds to a saturation concentration of 4.0 nmol L$^{-1}$. The atmospheric background signature of methane of -45 ‰ corresponds to the atmospheric $\delta^{13}$C-

CH$_4$ value (−47 ‰, Quay et al., 1991) corrected by the kinetic isotopic fractionation effect (Happell et al., 1995). In addition, methane was enriched in $^{13}$C, compared to the atmospheric background signature (Fig. 6b). The deviation in the $\delta^{13}$C-CH$_4$ values from this background value reflects the influence of sea ice methane-release.

The surface waters in this region are expected to be under-saturated due to increased solubility capacity inferred by cooling and freshening of their source waters (Damm et al., 2018). Conspicuously, we observe mainly super-saturation in seawater. The

estimated enhancement of the solubility capacity of about 10 %, is in line with the long-term cooling and freshening effect on the Atlantic waters forming the PSW (Rudels et al., 2000). It is unclear if the surface waters observed at our drift site was formed





remotely in the Eurasian Basin and returning with the TDS towards the Fram Strait (Damm et al., 2018; Rudels, 2012), or if it was formed more locally over the Yermak Plateau or in the adjacent Sofia Deep. Nevertheless, the final effect of cooling AW from its original characteristics by the major inflow region ($\theta > 3°$ C, S_P > 35, Orvik and Niiler, 2002) to the T/S observed at the drift site would be the same; with a major part of the solubility enhancement being due to the cooling (9 %) while the rest is due to the still modest freshening at this time of the year. Based on our data, we suggest methane release from sea ice as source of the observed excess. In the following, we discuss these findings and the relation of the pronounced variations in saturation levels to the characteristics in the underlying waters.

**4.2.1 Methane excess in PSW by release from sea ice when melt starts**

Even though the area of our drift was relatively small (cf. Fig. 1b), there were pronounced differences in methane excess in the seawater samples (Fig. 7a). As stated above, it is known that when ice starts to melt, brine is first released (Eicken, 2002) and concurrently methane dissolved in brine, also enters the ocean (Damm et al., 2015b). The T/S profiles at most of the methane sampling stations, indicate the onset of the typical-for-spring heating from above and an associated freshening due to melting (Fig. 8). Even if not certain that all the warming/freshening may be attributed our own floe, it has indeed traversed patches of water with the potential to onset basal melt. In our ice core samples we found a $\delta^{13}$C signature of methane in brine enriched in $^{13}$C (Table 1). Enrichment in $^{13}$C, compared to the background value, was also observed in the seawater samples (Fig. 6b), and we therefore conclude that brine release had occurred. E.g. at stn24-1 a warmer and fresher ($\Delta T \sim 0.03°$ C, $\Delta S \sim 0.01$ g kg$^{-1}$) layer down to about 40 m was found to contain high methane saturation level reflecting both sea-ice melt and sea-ice methane release (Fig. 7a and 7c). An early melt stage (initiated basal melt) would be indicated by a low degree of dilution of the released methane since only small amounts of meltwater are available at first. This is indeed observed in the methane super-saturation (up to 20 %), related to the saturation capacity at the YP. Additionally, the varied $\delta^{13}$C-CH$_4$ values detected in seawater corroborates the above assumption (Fig. 6b, Fig. 9).

In summary, the excess of methane in the surface water clearly point to sea-ice-sourced and early melt events as most important factors for methane release.

**4.2.2 Variability on methane saturation levels in PSW by oceanographic processes.**

During our drift, the stations were affected by a varying degree of ice melt, i.e. a varying degree of warming/freshening, with implications on the stratification and the potential to preserve the released methane (Fig. 7c and 7d). Methane released from sea ice into a shallow layer limited by strong/stable stratification directly underneath would be mixed/diluted to a certain degree but nevertheless be relatively well preserved, as the super-saturation observed at stn29-8 and 30-2 (at 2 m). This assumption is corroborated by values more enriched in $^{13}$C, coincident with these "hot-spots" of methane underneath the ice. While methane released into a deep weakly stratified layer would spread deeper, as detected in stn22-2, where the deepest calculated "winter mixed layer" was detected, reflecting well-blended waters.

By the stations with large melt, the methane excess seems to be preserved/sustained in the melt water layer by help of the increased stratification underneath this; down to about 40 m depth at stn24-1 and down to 20 m depth at stn27-6 (Fig. 7b and c). However, the conditions at stn27-6 seem more complex than at the "normal-looking" stn24-1, with a relatively warm and salty layer interleaving at 25-40 m depth reflected in a decreased methane saturation level at these depths (Fig. 8).

Further, the spatial variation of super-saturation levels could be also due to differences in the advection speed and direction of the drifting floe relative to the traversed waters. During high spatial and temporal coherence (ice and water make a joint journey), the contact time between the floe and the water is prolonged, and we imagine a well-coupled system of sea ice and the water underneath may be established. The advection of ice and water is summarized by the vectors in Fig. 7e, still with the prominent tidal signal contained. Both the average and maximum speed of water was similar at the two depths (0.03/0.20 ms$^{-1}$ at 10 m and 0.02/0.25 ms$^{-1}$ at 50 m, respectively) and lower than for the ice floe (0.09/0.30 ms$^{-1}$). Nevertheless, it is evident that the interplay between ice and water motion is very variable and shaped both by atmospheric (winds) and oceanic forcing (tides). Judging from this time lapse, the more joint journey is made during the last days of the drift, when both the speed and direction of the ice and water seem to be well aligned over the sloping topography of the YP southeastern flank. However, it is during the first part of the drift (over shallower grounds) with the more "back-and-forth" rotational nature of the ice and water movements which seems to have contributed to the fact that we can detect a methane super-saturation, sustained in the waters around our drift. In contrast, the more advecting nature of the last part of the drift along the slope may have prevented us from fully catching the methane signal, resulting



in the observed lower methane saturation levels (e.g. mostly at stations 31-3 and 32-5). According the drift direction, one would expect to see a similar pattern all through stations 29-8 to 32-5, but between 40-60 m at stations 29-8 and 30-2, waters with slightly different characteristics (colder, saltier) and more methane were observed (Fig. 8). Summarized, the fate of sea ice methane-sourced in surface water is subjected to the spatial and temporal coherence of the coupled sea ice- PSW system during the drift as well as how stratification acts to retain this excess in the surface waters.

## 5 Outlook/conclusion

The type and structure of Arctic sea ice affects the capacity for methane storage (Fig. 9). Our study gives evidences that ridged/rafted sea ice structures provide an interface where methane oxidation occurs during the journey in the Transpolar Drift Stream (TDS), eventually acting as a sink for methane. Further research should consider rate measurements of methane oxidation mainly in ridged/rafted ice structures to determine the impact of this process on a long-term scale.

We suggest that sea ice methane-released into the ocean, and in this case into the PSW, is the favored pathway in early spring. However, the sea ice-to-air-emissions still need to be considered, especially in FYI, due to its susceptibility to become permeable faster and thus encourage fluxes across the sea ice/air interface. Ongoing shifts from thick and complex to thin ice transported by the TDS is likely to cause changes on the final fate of methane.

Our study suggests that the excess of methane in PSW is sea ice-sourced and that the ongoing ice melt process influences this excess, both temporally and spatially. The degree of ice melt impacts the stratification and, hence, the potential for the sea-ice released methane to be retained in the surface waters. The final fate of the methane (excess) thereafter depends on to which extent it is diluted by additional meltwater. The general shift towards longer summer melt, has the potential to increase the amount of meltwater in the upper water column, which may inhibit the efflux into the atmosphere. Further studies are needed to investigate the relevance of more ice-free waters in summer and the potential to increase the ocean-air interactions for the methane emissions into the atmosphere.

Under the scenario of warmer AW inflow waters, the sink capacity of the surface waters for sea ice released methane may be potentially enhanced, hence with modifications of the methane pathways in the Arctic Ocean. Especially vulnerable for such changes are the areas beyond the current inflow area in the Eurasian basin, where the effect of the "Atlantification" is expected to be enhanced (Polyakov et al., 2017).

Finally, we think that one important factor determining the methane saturation levels below the ice, is to which extent the ice and seawater makes a "joint journey" (Damm et al., 2018), i.e. how the speeds of the ice and water relate to each other as well as how stratification acts to retain the methane signal in the surface layers. Extended analyses, and precise process modelling considering the entire complex sea ice-ocean (and atmosphere) system, is needed to improve our ability to predict the consequences of the methane source-sink balance modifications in the Arctic Ocean.

Author contribution

J.V. wrote the manuscript. J.V. and E.D. carried out the geochemical analyses and A.N., the oceanographic analysis. All authors contributed to the interpretation of the data, the manuscript and the figures.

Data availability.

Data will be available at www.pangea.de

Acknowledgements. We sincerely acknowledge the support of the captain and crew of R/V *Polarstern* cruise ARK-XXXI/1.1 for their professional support at sea. This study was funded through the Alfred-Wegener-Institute, Helmhotz-Zentrum für Polar und Meeresforschung. Expedition grant number AWI_PS106_00. J.V. received a scholarship from the National Agency for Research and Development (ANID)/Scholarship Program/Becas de Doctorado con acuerdo bilateral en el extranjero CONICYT-DAAD/2016–62150023. DAAD reference number 91609942.

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



**Table 1: Characteristics of the ice stations showing the information about the stations name, date of sampling during the ice drift in 2017, the ice thickness, snow thickness and the stable isotopic composition of methane ($\delta^{13}$C-CH₄) values in brine sampled following the "sackhole" method (e.g. Gleitz et al., 1995).**

| Station | Date of sampling | Ice thickness (cm) | Snow thickness (cm) | $\delta^{13}$C-CH$_4$ (‰) in brine | | |
|---|---|---|---|---|---|---|
| C3b | 4-Jun | 90 | 2 | | | |
| C1 | 4-Jun | 160 | 2 | | | |
| C4 | 5-Jun | 237 | 2 | | | |
| C6 | 8-Jun | 271 | 13 | | | |
| C7 | 9-Jun | 135 | 6 | | | |
| C8 | 10-Jun | 220 | 34 | -36.26 | | |
| C9 | 11-Jun | 179 | 44 | -36.43 | -37.38 | -37.06 |
| C10 | 12-Jun | 210 | 0 | -39.70 | | |
| C11 | 14-Jun | 278 | 90 | -43.57 | | |



(a)

(b)

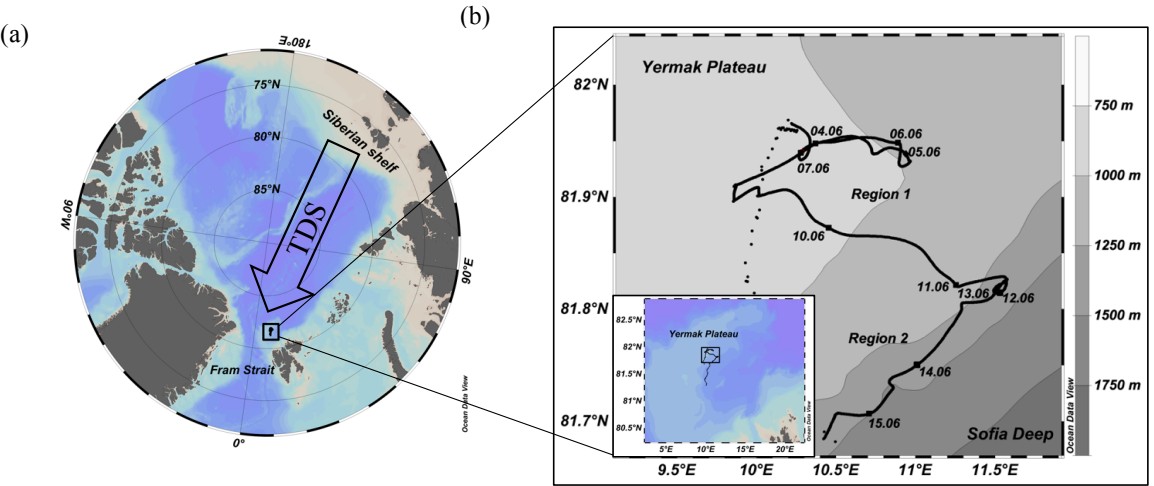

(c)

(d)

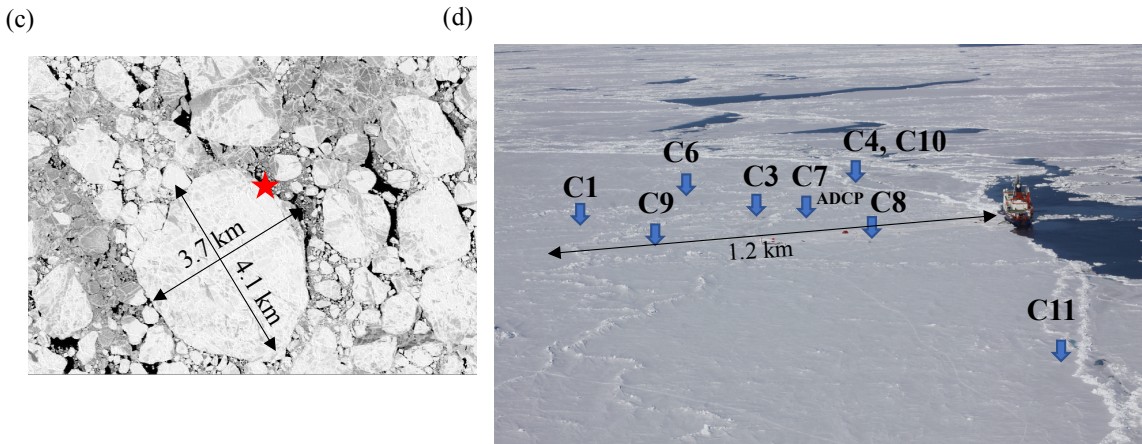

**Figure 1: (a) The location of the PS106.1 study area north of Svalbard (black rectangle). The arrow shows the general direction of the Transpolar Drift Stream (TDS). (b) The drift track of the ice camp between 4-15 June 2017, overlaid on the bathymetry of the region. (c) Satellite image of the ice floe serving as our drifting platform. The star shows the location of the RV *Polarstern*. (d) Positions of the ice coring stations. Note that C4 and C10 were taken at the same location, but on different dates. Photo credit (Fig. 1d): Giulia Castellani.**

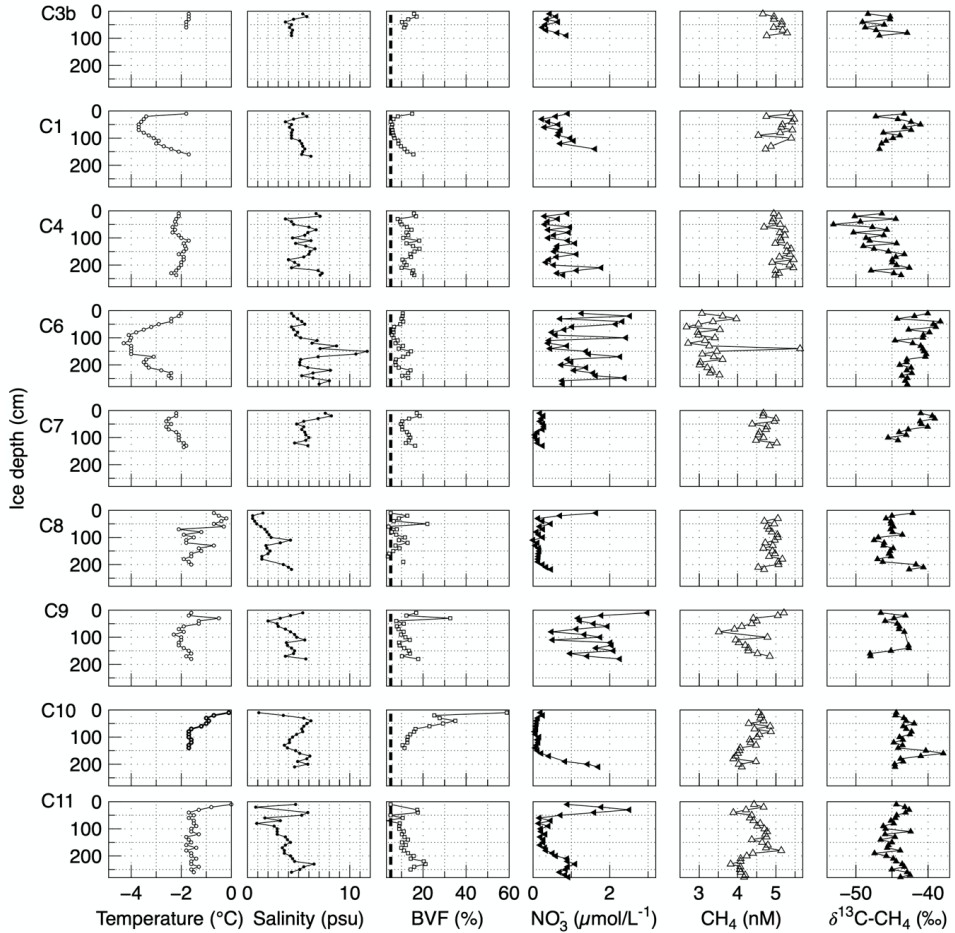

**Figure 2: Temperature, salinity, brine volume fraction (BVF), nitrate (NO₃⁻), methane concentration (CH₄) and the isotopic composition of methane (δ¹³C-CH₄) from sea ice cores taken across the ice floe. Black dashed line in BVF indicates when values are <5 %, corresponding to impermeable sea ice (Golden et al., 1998).**





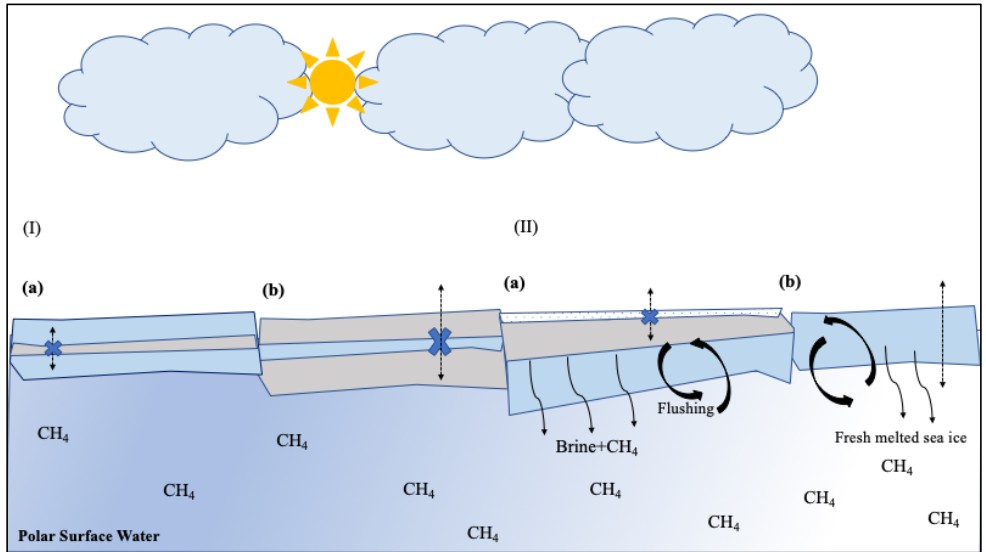

**Figure 3: Potential pathways of methane in sea ice with varying impermeable (indicated in grey) and permeable sections (in blue), i.e. winter (I) and spring (II) conditions. I (a) Relics of the initial methane signal (source) entrapped in impermeable ice. Impermeable intermediate sea ice layers, act as a barrier for the upward/downward transport of methane. (b) Residual methane signal after methane oxidation occurred in permeable sea ice enclosed by impermeable ice layers (see Fig. 4). II (a) When basal melt starts, downward brine transport initiates release of dissolved methane. Flushing events trigger methane released into the ocean. (see chapter 4.1.3). (b) Un-restricted migration of methane in permeable sea ice. Ongoing sea ice melt, freshwater from melted sea ice is released into the water underneath, resulting on a "freshwater pool", where methane remains sustained during the melt season.**



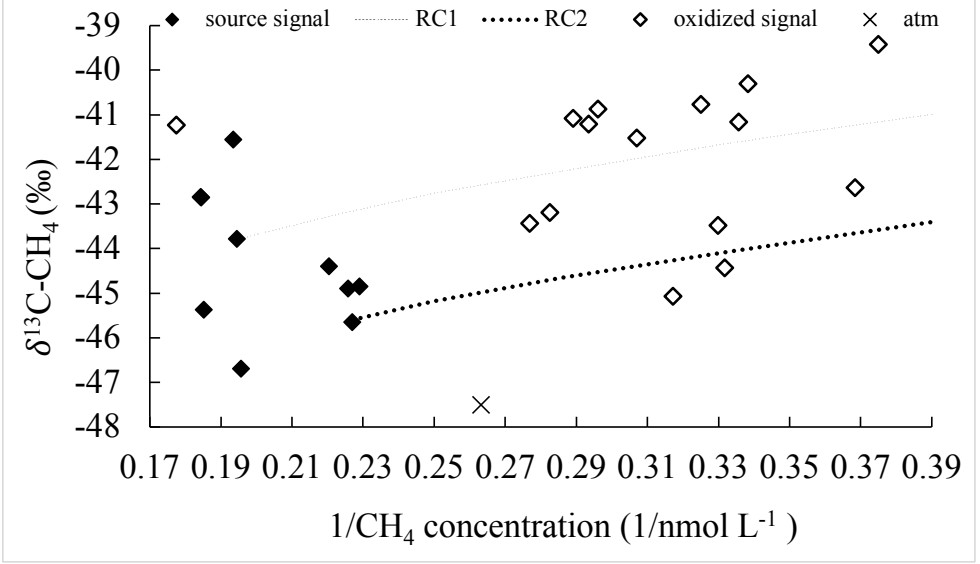

**Figure 4: The reciprocal of methane concentration vs. the $\delta^{13}$C-CH$_4$ in sea ice. Methane enclosed in impermeable layers (black diamonds) deviates from the atmospheric background value (cross) and reflects the initial methane (source) signal being unchanged stored in impermeable sea ice during the drift. The source signal also represents the starting point for the calculation of potential methane consumption. The residual methane (open diamonds), i.e. a pool with less methane concentration, but more enriched in $^{13}$C, is formed by methane consumption in a permeable layer enclosed by impermeable rafted ice. Two Rayleigh curves (RC1/dashed line and RC2/dotted line) have been calculated with two different initial isotopic signatures (-44 and -46 ‰), respectively, $\alpha$ is 1.004 (see 4.1.1 and 4.1.2).**





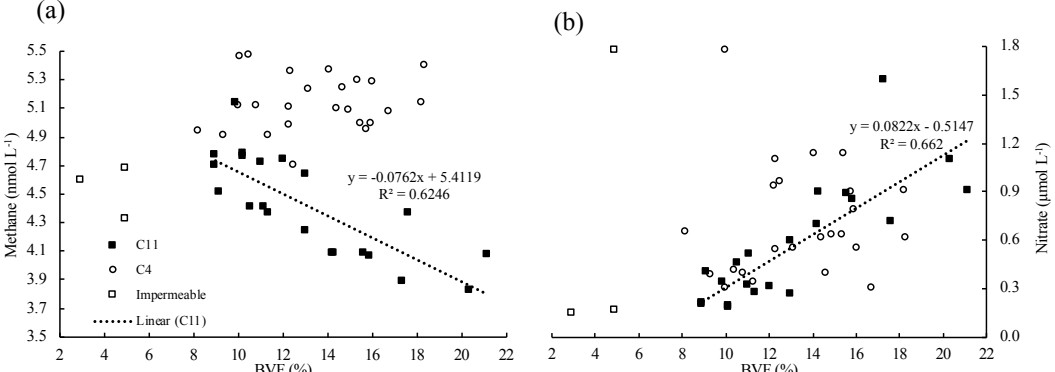

**Figure 5: (a) Methane and (b) nitrate concentration vs. brine volume fraction (BVF) at C11 and C4, respectively. In C11, methane is inversely correlated ($R^2$=0.62) while nitrate is correlated ($R^2$=0.66) with increasing BVF. This trend shows that methane is released from ice while nitrate is taken up. In station C4, correlation is missing when sea ice is fully permeable. C4 shows an homogenously distribution of methane as well as nitrate, and higher concentration than in C11, caused by seawater charged in methane and nitrate is flushed into the sea ice. The three outliers points (impermeable layers) have been removed for the correlations.**





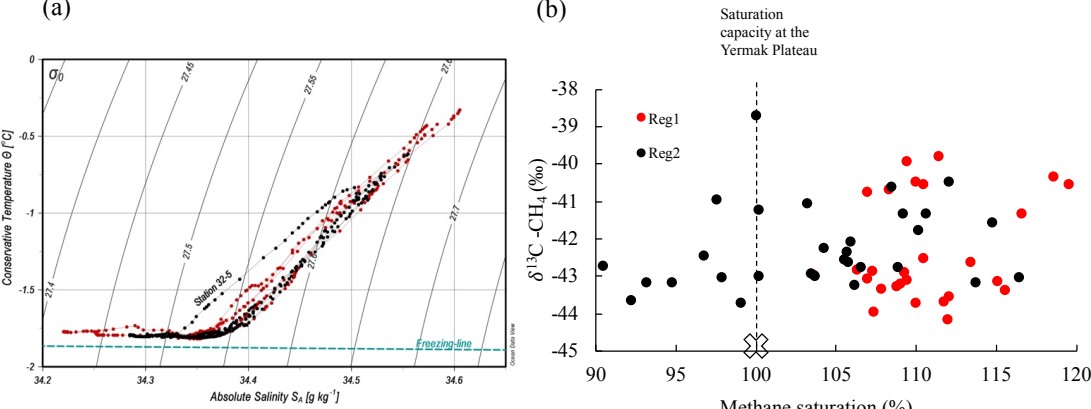

**Figure 6: (a) TS-diagram for the upper 100 m at the methane sampling stations during the ice drift. In red, the stations located over the Yermak Plateau (Region 1) and in black, over the Yermak Plateau eastern flanks (Region 2). The dashed line indicates the salinity dependence of the freezing temperature. (b) Methane saturation vs. the $\delta^{13}$C signature of methane in seawater. Red and black colors indicate the regions and the dashed line, the saturation capacity (100 %) at the Yermak Plateau (see section 4.2). The atmospheric background signature of methane of -45 ‰ is marked with a cross.**



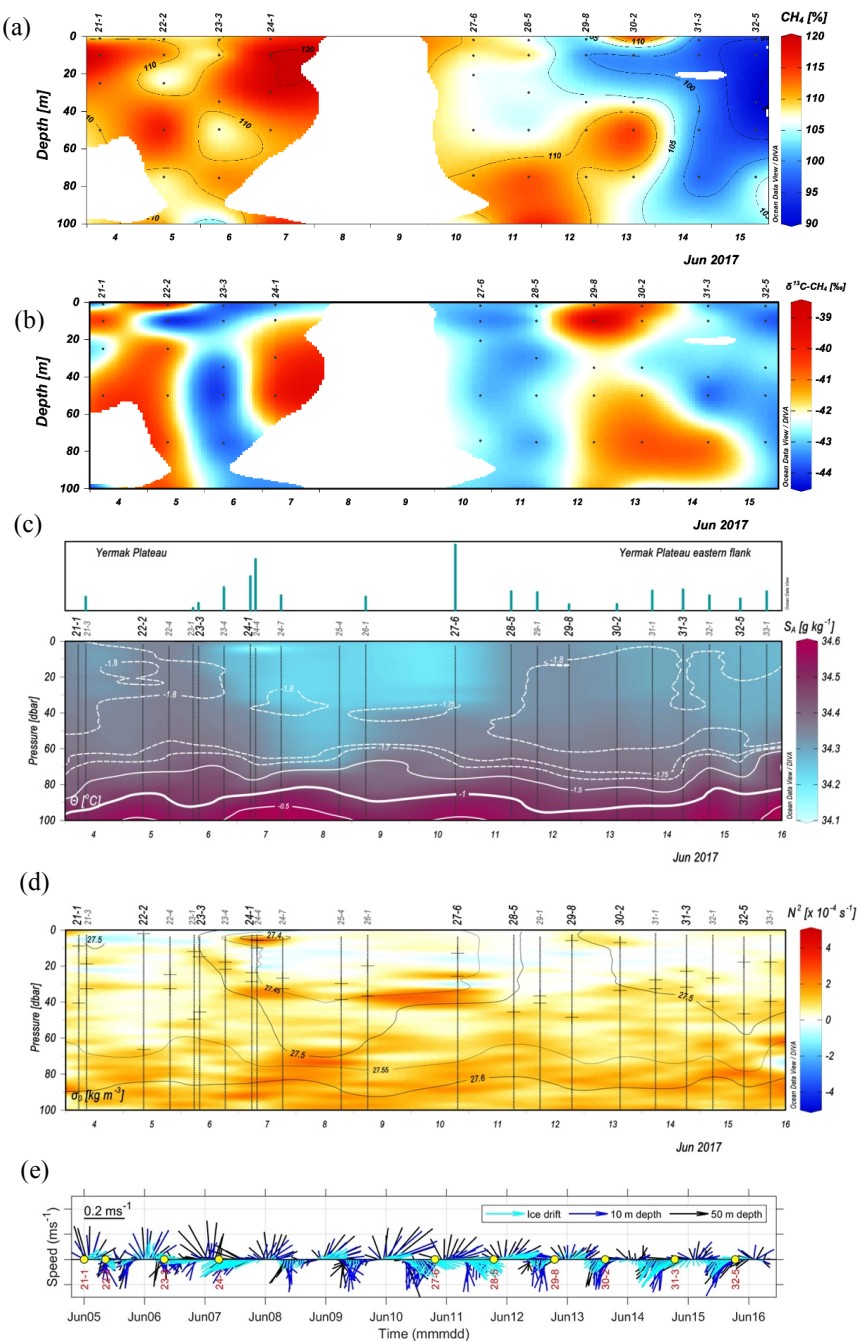

**Figure 7: Vertical distribution of several parameters in the upper 100 meters, during the entire drift 4–15 June, 2017; (a) the temporal development of methane saturation (color bar) overlaid with contours of saturation levels and (b) of the $\delta^{13}$C-signature of methane. (c) absolute salinity (g kg$^{-1}$) overlaid with isothermals of conservative temperature (° C) and (d) Brunt-Väisälä Frequency (10$^{-4}$ s$^{-1}$) overlaid with isopycnals of potential density anomaly (kg m$^{-3}$, 0 dbar ref. pressure). Positive $N^2$ values indicate stable stratification. Black horizontal bars indicate the mixed layer depths estimated for two different density thresholds (upper: 0.003 kg m$^{-3}$, lower: 0.01 kg m$^{-3}$, see 3.2). The labels on top of each panel denote the station numbers, with the methane-sampling stations in black. The**





**bars in the uppermost panel show the degree of ice melt at each station estimated from the T/S profiles following Peralta-Ferriz and Woodgate (2015). Their length scale has been omitted to emphasize this is only used as a qualitatively guidance this early in the melt season (melted ice ~ 0-10 cm). (e) Hourly vectors of the horizontal speed of the ice floe and the underlying waters, at 10 m and 50 m depth, respectively. The vector magnitudes follow the length scale in the upper left corner. The methane sampling stations are indicated with labeled yellow dots. Figures 7a-7d were made in ODV (Schlitzer, 2020) and 7e in Matlab R2018b.**





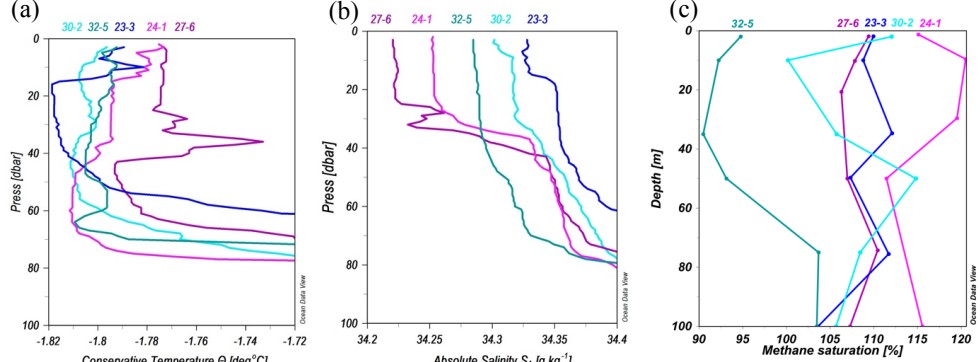

**Figure 8: (a) Conservative temperature (° C), (b) absolute salinity (g kg⁻¹) and (c) methane saturation (%) profiles for some of the sites with warmer and fresher waters in the topmost layers, indicating the onset of the seasonal ice melt to various degree. The stations are indicated by the labels on top of each panel.**



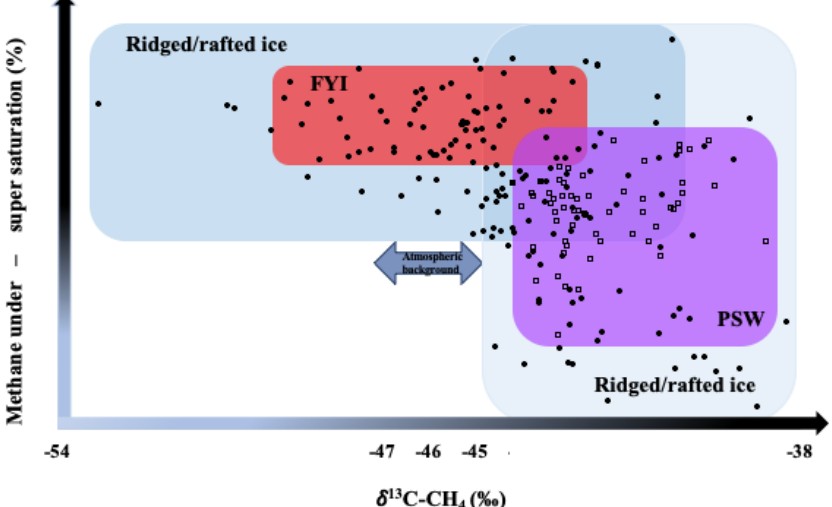

**Figure 9: Variability in the methane inventories within different ice types on our ice floe and PSW underneath. The arrow shows the deviation to the atmospheric background signature.**