# Peer review of "Methane cycling within sea ice; results from drifting ice during late spring, north of Svalbard"

_The Cryosphere, 2020_

## Referee Comment (RC1) · Anonymous Referee #1 · 10 Nov 2020

This manuscript presents a unique data set of observations of methane concentrations and isotopic composition in Arctic sea ice and ocean during late spring in the Eurasian Basin. Several types of sea ice were sampled allowing to explore the impact of sea ice structure and age on storage and release of methane in the environment. This is an area of active research with potential for strong implications for the climate system. The theory presented here is that in the observed conditions (late spring Eurasian Basin), the predominant methane pathway is from sea ice into the ocean as opposed to sea ice to the atmosphere. Also discussed is the impact of ocean dynamics such as current and stratification on the storage and further pathways of methane into the system.

The quality of the English is mostly ok but lacks fluency and needs some attention. The introduction covers well the topic. The Material and Methods section is excellent.

The Results section should be less repetitive and less descriptive. The Discussion is interesting even if it sometimes sounds speculative; this might be inherent to the topic of methane pathways in Arctic sea ice and ocean which is still in its infancy. The Summary section is good and the figures are overall quite clear.

Major comments:

- Results: Most of the results are direct descriptions of the figures without any further take or analysis. It is very repetitive and sometimes hard to link to the further steps you take in the Discussions. It might be useful to move some of the discussion results in the Result section.

- Discussion and conclusions: what are your main findings or take-home message? At the moment it is not obvious. For example, in the abstract the last sentence states 'We point to sea ice as a potential source of methane'. This sounds very speculative with 'we point' rather than 'we find'; and with 'a potential'. Either be more assertive about what you have found, or if you must stay speculative, then suggest more alternative theories that explain your observations. For example, you never mention the potential of the local sediments as sources of methane in the water column, for example nearby or on the Yermak Plateau? Why could this not be the case? Please try to better highlight your key findings throughout the manuscript (abstract, discussion and conclusions).

Minor comments:

- Some English grammatical and formulation issues, as well as missing words and typos. Please read and check carefully before resubmission.

This manuscript presents original and novel data, and the purpose of the work is

clearly articulated. The results section needs a re-write and the discussion needs to be strengthened to be less speculative and better highlight the key findings. I recommend the manuscript to be reconsidered after major revisions and look forward to seeing a revised version.

Individual comments

1.Introduction:

- L25-27: Add somewhere that you are talking about enhanced methane emissions from the ocean to the atmosphere.

- L27: 'Because the Arctic holds large natural sources of this highly potent...' again, do you mean the 'Arctic Ocean'?? or sea floor or sediments?

- L37: 'sea ice charged with methane' Consider using 'sea ice loaded with' or another term?

- L39: 'during the last years' please add if you can a time period here to better indicate what you mean by 'the last years'.

2. Material and methods:

- Section 2.2: excellent

- Section 2.3: excellent

- Section 2.4: excellent

3. Result:

- Table 1: For clarity, could you add '-' when you don't have values for the isotopic composition? I expect that for station C9, the 3 values of isotopic composition is 3 different estimates? Can you add that information in the Table caption?

СЗ

- L122: 'has an age of 1 to 3 years, respectively' What do you mean by respectively there?

- Section 3.11 and 3.12: These 2 sections are rather tedious reading; they very carefully describe each panel of Fig2 one by one. There might be a way to pull the numbers together in a way that builds on from Fig2.

- Line 196: 'at 90-100 m depth (Fig. 6a and 7a).' You refer to Fig 6 and 7 after Fig 2, without having mentioned Fig 3-5. You will have to reorder the figures to match the order you refer to them.

- Figure 7: There are 6 subplots but you only label 5 of them (a-e). The odd one out is the 3rd from the top, which I think is the ice melt estimates? Please add a subplot letter and clarify the figure caption.

4. Discussion:

- Fig.3: Very nice figure. We can guess most of the media but they should still be annotated: Atmosphere, ocean, sea ice, snow? In (IIa), what's the white section with blue dots? Why the gradual change in color of the blue ocean? And do all the CH4 annotation in the ocean indicate concentrations? If so, make it more obvious. Also add a definition of the black arrows in the caption.

- L278-279: 'With changes in sea ice dynamics, more of this complex ice structures may be formed, which in turn may promote changes on the methane cycling within sea ice.': here you mean to discuss implications for the future Arctic but its not obvious. Please rephrase.

- L343-344: 'In summary, the excess of methane in the surface water clearly point to sea-ice-sourced and early melt events as most important factors for methane release.' . This seems like too strong a statement considering the evidence you have presented.

- Section 4.2.2: some attention needs to be given to the English in this section, with many language choices that are not English based. Eg: 'the more joint journey is

made...', 'According the drift direction, one would'

5. Outlook/conclusion

- L379: 'We suggest that sea ice methane-released into the ocean, and in this case into the PSW, is the favored pathway in early spring.' Do you mean anywhere? In the whole Arctic? In this region only? Please add details.'

- L385: 'The final fate of the methane (excess) thereafter depends on to which extent it is diluted by additional meltwater.' What about the dilution by ocean mixing, currents, tides etc..? You don't mention the role of stratification here.

- L390-393: You mention warmer waters and Atlantification, Atlantification also changes the vertical ocean stratification in the region. If stratification was to increase, then methane released in surface waters could be trapped close to the surface during summer, leading to potentially increased exchanges with the atmosphere (and transfers into the atmosphere). If stratification was to decrease, methane could spread deeper into the ocean.

- L394-398: The overall transfer of methane from sea ice to the ocean stays the same, whether the ice and ocean 'travel' together or not. But changes in how far from the source, the methane is released into the ocean and atmosphere. This you don't mention here. There is also the acceleration of sea ice drift in the Arctic which means that sea ice rich in methane that formed on the Siberian shelves, is now potentially drifting further out with the TPD towards Fram Strait before melting and changing and therefore before releasing its methane.

- L396-398: Process modelling studies are a great idea.

Figure 1: Make it clearer which part of the drift is in Region 1 and which part in Region 2.

Figure 9: Add 'Within ridges / rafted sea ice' for FYI and 'Under ridged / rafted sea ice' for PSW on the figure. The arrow for the atmospheric background signature is not

great. Could you not instead have a dot, or create another color rectangle to represent standard local atmospheric ranges?

---

## Editor Comment (EC1) · Christian Haas (Editor) · 12 Dec 2020

Dear authors,

Unfortunately we have been unable to receive a second reviewer comment in any acceptable time, even though we already extended the deadline six times, for a total period of 7 weeks. As there were no other public comments either, I provide a short editor evaluation here to be able to move the manuscript forward, noting that I am not an expert on the specific topic.

However, as my own, initial impression of your manuscript was quite positive, and as the one review we have is generally positive as well, and has mainly formal concerns, I will now close the discussion and move forward with your response and revision of the

work. Please take the comments of the reviewer carefully into account, and improve your writing and balanced discussion as suggested in there. Once you have submitted your comments, and then the revisions, we will send the manuscript out for another review again. Please note that you will first be asked to provide short, high level replies that show that you can sufficiently improve your manuscript, and that detailed replies and comments are only required once you submit your revised manuscript.

Thank you for your patience, and hopefully we can now move on swiftly. Best regards Christian Haas

---

## Author Comment (AC1) · 21 Dec 2020

Dear Editor, We send answers to comments made by Reviewer#1 as suggested, which show that we are able to sufficiently improve our manuscript. Sincerely, Josefa Verdugo

Answers to comments made by Reviewer#1

Why methane released from e.g., sediments nearby or on the Yermak Plauteu is unlikely to be a source for supersaturations in surface waters?

Answer:

We will add this information to discussion: 4.2. Methane excess in Polar Surface Water (PSW) by release from sea ice when melt starts.

Significant methane released from sediments is reported to occur in areas of West of Svalbard (Sahling et al., 2014; Smith et al., 2014; Westbrook et al., 2009). In addition, methane bubble release was also documented at the deep sea Haakon Mosby Mud Volcano, on the SW Barents Sea slope (Sauter et al., 2006). However, most of this methane released by sediments are laterally transported within the deeper ocean and does not reach the surface waters (Damm et al., 2005; Damm and Budéus, 2003; Graves et al., 2015). In addition, while transported by ocean currents, dissolved methane is partly oxidized by microbes and partly diluted by mixing with background methane transported by ocean currents (Mau et al., 2017). Additionally, the sea ice drifted from East to West, while the sediment sources are west of Svalbard, i.e., the source area is localized in South-West direction of our study area. Hence, sea ice formation and methane uptake occurred in a region far away from these known sources.

- L278-279: 'With changes in sea ice dynamics, more of this complex ice structures may be formed, which in turn may promote changes on the methane cycling within sea ice.': here you mean to discuss implications for the future Arctic but its not obvious. Please rephrase.

Answer:

Will be changed to:

In response to Arctic amplification of global warming, thinner sea ice is expected to occur and, more of these micro-niches formed during ice ridging may lead to favored methane oxidation therein. Under these circumstances, the methane pathways can be modified, i.e., sea ice may be considered as a sink for methane. In addition, with an accelerated sea ice transport, methane up-taken in sea ice will be transported to remote areas, and released in surface waters of regions up to now not affected by methane excess.

- L379: 'We suggest that sea ice methane-released into the ocean, and in this case into the PSW, is the favored pathway in early spring.' Do you mean anywhere? In the

whole Arctic? In this region only? Please add details.'

Answer:

This information will be included in outlook/discussion

We suggest methane release from sea ice into the meltwater layer as yet unconsidered pathway mainly in early spring when the top of sea ice is still impermeably, but basal melt started. Further investigations are needed to estimate the amount of methane released into the atmosphere by sea ice to air flux, compared to the amount released by brine rejection into the marine environment. All methane pathways should be considered for the parameterization of process modelling studies. However, detailed process studies of sea ice formation on different Arctic shelves are crucial to validate the importance of methane uptake during ice formation.

- L385: 'The final fate of the methane (excess) thereafter depends on to which extent it is diluted by additional meltwater.' What about the dilution by ocean mixing, currents, tides etc..? You don't mention the role of stratification here.

Answer:

This information will be included in outlook/conclusion

Our study suggests that the excess of methane in PSW, at this time of the year, is sea ice-sourced and that the ongoing ice melt process influences this excess, both temporally and spatially. The degree of ice melt impacts both the actual freshwater content and the stratification and, hence, the potential for the sea-ice released methane to be retained in the meltwater layer. The sea to air flux is inhibited by the formation of the meltwater layer, and increasingly so during its seasonal development (i.e., freshening and warming) and its deepening through wind induced mixing. The methane excess trapped within this layer is subjected to this freshwater discharge and diluted, in a latest stage of melt. Mixing may also be inferred by ocean currents, and their interplay with the drifting ice. At occasions of strong winds the shear (i.e., the velocity difference

between the two media) will be particularly large and give rise to enhanced turbulence in the upper water layers affected by the motion (and speed) of the ice. The Yermak Plateau area is, in general, identified as a region of large tidal variability and enhanced mixing rates (Fer et al., 2015; Meyer et al., 2017). These tidal mixing mechanisms are mainly manifested at deeper levels due to the interference between the tidal flow and the very variable bottom topography. Since our study focuses on the upper 100 m, the effect of tidal mixing should be minor. Whichever mixing mechanisms were at play during our study, all CTD profiles during the drift showed that the upper 100 m consistently where characterized as PSW, meaning our samples were taken above the depths with substantial stratification (i.e., the main pycnocline, between surface waters and AW-influenced waters) and above the depths were waters showed an increasing influence of Atlantic-origin waters.

- L390-393: You mention warmer waters and Atlantification, Atlantification also changes the vertical ocean stratification in the region. If stratification was to increase, then methane released in surface waters could be trapped close to the surface during summer, leading to potentially increased exchanges with the atmosphere (and transfers into the atmosphere). If stratification was to decrease, methane could spread deeper into the ocean.

Answer:

It will be included in outlook/conclusion

For the potential long-term effect, we relate to the effect of an increased ocean heat content leading to enhanced ice melt and, hence, more freshwater discharged into the surface layer. A fresher (and perhaps thicker) surface layer 'cap' than today would imply stronger stratification efficiently inhibiting the exchange between the atmosphere and the subsurface ocean layers. Any methane excess in the waters below this 'cap' would thus be disconnected from the atmosphere and remain preserved. Within the surface layer itself, however, a larger amount of freshwater would potentially lead to

an increased dilution effect. Hence, Atlantification finally contribute to disconnect the meltwater layer and a potential methane excess therein from the atmosphere. Further, an enhanced deep winter convection may lead to a weakened stratification, and downwards transport of methane formerly preserved in meltwater layers into the entire PSW. Hence, we suggest that Atlantification most likely will enhance the sink capacity of the PSW for methane, either by dilution in the PSW itself or by mixing into the deeper ocean.

- L394-398: The overall transfer of methane from sea ice to the ocean stays the same, whether the ice and ocean 'travel' together or not. But changes in how far from the source, the methane is released into the ocean and atmosphere. This you don't mention here. There is also the acceleration of sea ice drift in the Arctic which means that sea ice rich in methane that formed on the Siberian shelves, is now potentially drifting further out with the TPD towards Fram Strait before melting and changing and therefore before releasing its methane.

Answer:

It will be included in outlook/conclusion

The overall transfer stays the same, but there is a difference in terms of the source-sink balance if methane is transported within sea ice or in seawater underneath when released. It is not yet clear which process contributes to a larger amount to the release of methane from sea ice, either the brine release during freezing in winter or during melting in spring. Both processes have to be considered and the amount of methane has to be quantified, before we are able to discuss the potential effect of the acceleration of the sea ice drift. One of the main results of our study is that we show for the first time that methane oxidation occurs in certain layers of complex sea ice structures. Under this circumstance, sea ice might act as a sink for methane. A faster sea ice drift (Spreen et al., 2011) resulting from a thinning ice cover on one side may reduce the time for methane to be oxidized within the ice, leading to changes in the methane

pathways. On the other hand thinner ice breaks up more easily by winds and waves resulting in the formation of more leads (open waters). This, in turn, may enhance the exchange of methane between the ocean and atmosphere (Kort et al., 2012).

References: Damm, E. and Budéus, G.: Fate of vent-derived methane in seawater above the Håkon Mosby mud volcano (Norwegian Sea), Mar. Chem., 82(1–2), 1–11, doi:10.1016/S0304-4203(03)00031-8, 2003. Damm, E., Mackensen, A., Budéus, G., Faber, E. and Hanfland, C.: Pathways of methane in seawater: Plume spreading in an Arctic shelf environment (SW-Spitsbergen), Cont. Shelf Res., 25(12–13), 1453–1472, doi:10.1016/j.csr.2005.03.003, 2005. Fer, I., Müller, M. and Peterson, A. K.: Tidal forcing, energetics, and mixing near the Yermak Plateau, Ocean Sci., 11(2), 287–304, doi:10.5194/os-11-287-2015, 2015. Graves, C. A., Steinle, L., Rehder, G., Niemann, H., Connelly, D. P., Lowry, D., Fisher, R. E., Stott, A. W., Sahling, H. and James, R. H.: Fluxes and fate of dissolved methane released at the seafloor at the landward limit of the gas hydrate stability zone offshore western Svalbard, J. Geophys. Res. Ocean., 120(9), 6185–6201, doi:10.1002/2015JC011084, 2015. Kort, E. a., Wofsy, S. C., Daube, B. C., Diao, M., Elkins, J. W., Gao, R. S., Hintsa, E. J., Hurst, D. F., Jimenez, R., Moore, F. L., Spackman, J. R. and Zondlo, M. a.: Atmospheric observations of Arctic Ocean methane emissions up to 82° north, Nat. Geosci., 5(5), 318–321, doi:10.1038/ngeo1452, 2012. Mau, S., Römer, M., Torres, M. E., Bussmann, I., Pape, T., Damm, E., Geprägs, P., Wintersteller, P., Hsu, C.-W., Loher, M. and Bohrmann, G.: Widespread methane seepage along the continental margin off Svalbard - from Bjørnøya to Kongsfjorden, Sci. Rep., 7(February), 42997, doi:10.1038/srep42997, 2017. Meyer, A., Sundfjord, A., Fer, I., Provost, C., Villacieros Robineau, N., Koenig, Z., Onarheim, I. H., Smedsrud, L. H., Duarte, P., Dodd, P. A., Graham, R. M., Schmidtko, S. and Kauko, H. M.: Winter to summer oceanographic observations in the Arctic Ocean north of Svalbard, J. Geophys. Res. Ocean., 122(8), 6218–6237, doi:10.1002/2016JC012391, 2017. Sahling, H., Römer, M., Pape, T., Bergès, B., dos Santos Fereirra, C., Boelmann, J., Geprägs, P., Tomczyk, M., Nowald, N., Dimmler, W., Schroedter, L., Glockzin, M. and Bohrmann, G.: Gas

emissions at the continental margin west of Svalbard: mapping, sampling, and quantification, Biogeosciences, 11(21), 6029–6046, doi:10.5194/bg-11-6029-2014, 2014. Sauter, E. J., Muyakshin, S. I., Charlou, J.-L., Schlüter, M., Boetius, A., Jerosch, K., Damm, E., Foucher, J.-P. and Klages, M.: Methane discharge from a deep-sea submarine mud volcano into the upper water column by gas hydrate-coated methane bubbles, Earth Planet. Sci. Lett., 243(3–4), 354–365, doi:10.1016/j.epsl.2006.01.041, 2006. Smith, A. J., Mienert, J., Bünz, S. and Greinert, J.: Thermogenic methane injection via bubble transport into the upper Arctic Ocean from the hydrate-charged Vestnesa Ridge, Svalbard, Geochemistry, Geophys. Geosystems, 15(5), 1945–1959, doi:10.1002/2013GC005179, 2014. Spreen, G., Kwok, R. and Menemenlis, D.: Trends in Arctic sea ice drift and role of wind forcing: 1992-2009, Geophys. Res. Lett., 38(19), n/a-n/a, doi:10.1029/2011GL048970, 2011. Westbrook, G. K., Thatcher, K. E., Rohling, E. J., Piotrowski, A. M., Pälike, H., Osborne, A. H., Nisbet, E. G., Minshull, T. A., Lanoisellé, M., James, R. H., Hühnerbach, V., Green, D., Fisher, R. E., Crocker, A. J., Chabert, A., Bolton, C., Beszczynska-Möller, A., Berndt, C. and Aquilina, A.: Escape of methane gas from the seabed along the West Spitsbergen continental margin, Geophys. Res. Lett., 36(15), 1–5, doi:10.1029/2009GL039191, 2009.

---

## Author Comment (AC2) · 3 Jan 2021

Dear Reviewer 1,

Thank you very much for your feedback. We greatly appreciate your helpful comments and suggestions to improve our manuscript.

With respect to your major comments made

- Results: Most of the results are direct descriptions of the figures without any further take or analysis. It is very repetitive and sometimes hard to link to the further steps you take in the Discussions. It might be useful to move some of the discussion results in the Result section.

[Figure]

The ice conditions of our floe were very heterogeneous, e.g., different snow thickness, ice thickness, ice types/structures, hence we are convinced that a detailed description of all ice cores will be very helpful. Especially for readers interested in comparison of ice cores from later studies with the ice cores presented in our study.

But to improve the overview, we will include the following text in section 3.1.

"The ice floe was formed by FYI and ridged/rafted ice. The ice thickness at the sampled stations varied between 90 and 280 cm, while snow thickness on top of the ice varied from 0 to 90 cm (Table 1). Of the nine ice cores sampled on the ice floe, eight were taken in the ridged/rafted site along a 1.2 km transect (Fig. 1d). Ridged and rafted ice can be especially relevant for the methane cycling, due to the fact that they remain more consolidated even in the summer season (in comparison to FYI) and thus allow us to investigate methane-related processes in certain layers of these ice structures. Vertical profiles of temperature, salinity, BVF, NO3-, CH4 concentration and the ïĄd'13C-CH4 for all ice cores are shown in Fig. 2 with additional information in Table 1. Following Golden et al. (1998), permeable ice for gas migration was classified for a BVF above 5 % (see methods). To highlight the spatial variability of the sea ice's physical and biogeochemical properties, we present detailed descriptions of each ice core. To highlight the spatial variability of the sea ice's physical and biogeochemical properties, we present detailed descriptions of each ice core".

- Discussion and conclusions: what are your main findings or take-home message? At the moment it is not obvious. For example, in the abstract the last sentence states 'We point to sea ice as a potential source of methane'. This sounds very speculative with 'we point' rather than 'we find'; and with 'a potential'. Either be more assertive about what you have found, or if you must stay speculative, then suggest more alternative theories that explain your observations.

In the abstract the following sentence: We point to sea ice as a potential source of methane" it will be changed to: "We suggest sea ice loaded with methane as a source

of methane for Polar Surface waters during early spring".

We will include the following take-home messages:

- Our study provides evidence that ridged/rafted sea ice structures create environments where methane oxidation occurs during the Transpolar Drift Stream, eventually acting as a sink for methane. - We propose methane release from sea ice into the meltwater layer as a pathway, mainly in early spring when basal melt is occurring and the top of sea ice loaded with methane is still impermeable. - The ongoing ice melt triggers the methane excess formed in PSW, both temporally and spatially. - The relative velocities of the ice and water and the impacts of stratification on methane signal retention in the surface waters are important factors for the detection of sea ice-induced methane excess in seawater underneath the ice.

-For example, you never mention the potential of the local sediments as sources of methane in the water column, for example nearby or on the Yermak Plateau? Why could this not be the case? Please try to better highlight your key findings throughout the manuscript (abstract, discussion and conclusions).

In section 4.2 "Dissolved methane in Polar Surface Water (PSW)" we will include an explanation why we ruled out the sediment sources for the methane excess in PSW as followed:

"Irrespective of the mixing mechanisms at play during our study, all CTD profiles during the drift showed that the upper 100 m were consistently characterized as PSW. This means that our samples were taken above the depths with substantial stratification and above waters that were influenced by Atlantic-origin. In addition, methane released from sediments in the region West of Svalbard (Sahling et al., 2014; Smith et al., 2014; Westbrook et al., 2009) are laterally transported in the deep ocean and do not reach the surface waters (Damm et al., 2005; Graves et al., 2015; Silyakova et al., 2020). Hence, the PSW in our study area remains unaffected by methane released from seafloor sources further south. Based on our data and the regional oceanographic conditions,

we suggest that methane release from sea ice is the source of the observed excess in PSW".

With respect to your minor comments made

- Some English grammatical and formulation issues, as well as missing words and typos. Please read and check carefully before re-submission.

We agree, we will check the English grammar and formulation in the revised version of the manuscript.

With respect to your individual comments made

1.Introduction:

- L25-27: Add somewhere that you are talking about enhanced methane emissions from the ocean to the atmosphere.

It will be changed to: "In particular, sea ice retreat may quickly induce enhanced methane (CH4) emissions from the surface ocean into the atmosphere due to the loss of its barrier function for sea-air gas exchange (Wahlstrom and Meier, 2014)".

- L27: 'Because the Arctic holds large natural sources of this highly potent...' again ,do you mean the 'Arctic Ocean'?? or sea floor or sediments?

We will include the following sentence in the introduction section: "Accordingly, the methane reservoir estimate in the East Siberian and Laptev Seas, ranges from 1.6 and 5.7 Gg CH4 in the seawater, varying with season and depending on the ice cover (Shakhova et al., 2005; McGuire et al., 2009)".

And the following sentence will be removed "Because the Arctic holds large natural sources of this highly potent...' again ,do you mean the 'Arctic Ocean".

- L37: 'sea ice charged with methane' Consider using 'sea ice loaded with' or another term?

It will be changed to: "sea ice loaded with methane"

- L39: 'during the last years' please add if you can a time period here to better indicate what you mean by 'the last years'.

It will be changed to: "Recent trends in sea ice transported by the TDS, show that sea ice structure has undergone substantial changes since the early 1980s, shifting from thicker multi-year ice (MYI) to thinner and more fragile first-year ice (FYI; Zamani et al., 2019; Hansen et al., 2013; Maslanik et al., 2011, 2007)".

3.Result:

- Table 1: For clarity, could you add '-' when you don't have values for the isotopic composition? I expect that for station C9, the 3 values of isotopic composition is 3 different estimates? Can you add that information in the Table caption?

The following sentence will be added to the table caption: "In station C9, three samples were taken, while in C8, C10 and C11, one sample. The rest ice cores stations, no brine samples were taken".

- L122: 'has an age of 1 to 3 years, respectively' What do you mean by respectively there?

It will be changed to: "Backward drift trajectories suggest that our floe originated in the Siberian Sea, while the age of the sea ice was estimated to be between 1 and 3 years (Wollenburg et al., 2020)".

- Section 3.11 and 3.12: These 2 sections are rather tedious reading; they very carefully describe each panel of Fig2 one by one. There might be a way to pull the numbers together in a way that builds on from Fig2.

Answer is given in the beginning of our reply to major comments.

- Line 196: 'at 90-100 m depth (Fig. 6a and 7a).' You refer to Fig 6 and 7 after Fig2, without having mentioned Fig 3-5. You will have to reorder the figures to match the

order you refer to them.

Figures will be re-ordered in the revised manuscript

- Figure 7: There are 6 subplots but you only label 5 of them (a-e). The odd one out is the 3rd from the top, which I think is the ice melt estimates? Please add a subplot letter and clarify the figure caption.

The subplot letter will be added, and in the caption the following sentence will bee included:

"(c) the bars on the panel show the degree of ice melt at each station estimated from the T/S profiles following Peralta-Ferriz and Woodgate (2015). Their length scale has been omitted to emphasize that this is only used as a qualitatively guidance of the early melt season (melted ice $\sim$ 0-10 cm)".

4. Discussion:

- Fig.3: Very nice figure. We can guess most of the media but they should still be annotated: Atmosphere, ocean, sea ice, snow? In (IIa), what's the white section with blue dots? Why the gradual change in color of the blue ocean? And do all the CH4 annotation in the ocean indicate concentrations? If so, make it more obvious. Also adda definition of the black arrows in the caption.

"Atmosphere, ocean, sea ice, snow", will be added in the cartoon.

The caption will be changed to:

"Figure 6: Potential pathways of methane in sea ice with varying impermeable (indicated in grey) and permeable sections (in white with blue dots), i.e. winter (I) and spring (II) conditions. I (a) Relicts of the initial methane signal (source) entrapped in impermeable ice. Impermeable intermediate sea ice layers, act as a barrier for the upward/downward transport of methane (black arrow overlaid by a blue cross). (b) Residual methane signal after methane oxidation occurred in permeable sea ice ("water pocket"), enclosed by impermeable ice layers (see Fig. 7). II (a) When basal melt starts and impermeable layers still on top with snow cover (white layer on top of the ice), downward brine transport initiates release of dissolved methane. Flushing events trigger methane released into the ocean. (see chapter 4.1.3). (b) Un-restricted migration of methane in permeable sea ice (black dotted arrow). Ongoing sea ice melt, freshwater from melted sea ice is released into the water underneath, resulting on a "meltwater layer", where methane remains sustained during early spring. Methane (CH4) annotation indicates concentration. Blue gradient in the ocean, reflects the increasing stratification during the seasonal evolution of the upper part of the WML into a fresh meltwater layer".

- L278-279: 'With changes in sea ice dynamics, more of this complex ice structures may be formed, which in turn may promote changes on the methane cycling within sea ice.': here you mean to discuss implications for the future Arctic but its not obvious. Please rephrase.

It will be changed to: "In response to Arctic amplification of global warming, thinner sea ice is expected to occur. Thus, an increased number of permeable pockets formed during ice ridging may lead to favored methane oxidation therein. Under these circumstances, we suggest that the methane pathways can be modified, i.e., sea ice may be considered as a sink for methane".

- L343-344: 'In summary, the excess of methane in the surface water clearly point to sea-ice-sourced and early melt events as most important factors for methane release.'. This seems like too strong a statement considering the evidence you have presented.

It will be changed to: "In summary, the methane excess in PSW, at this time of the year, is likely to be sea ice-sourced and the ongoing ice melt process influences this excess".

- Section 4.2.2: some attention needs to be given to the English in this section, with many language choices that are not English based. Eg: 'the more joint journey is

made...', 'According the drift direction, one would'

The English grammar will be checked for the revised manuscript

5. Outlook/conclusion

- L379: 'We suggest that sea ice methane-released into the ocean, and in this case into the PSW, is the favored pathway in early spring.' Do you mean anywhere? In the whole Arctic? In this region only? Please add details.'

It will be changed to:

"We propose methane release from sea ice into the meltwater layer as a pathway, mainly in early spring when basal melt is occurring and the top of sea ice loaded with methane is still impermeable. Studies are needed to estimate the amount of methane released into the atmosphere by the sea ice-to-air flux compared to the amount released by brine rejection into the marine environment".

- L385: 'The final fate of the methane (excess) thereafter depends on to which extent it is diluted by additional meltwater.' What about the dilution by ocean mixing, currents, tides etc..? You don't mention the role of stratification here.

It will be changed to:

"Our study suggests that the excess of methane in PSW during early spring, is sea ice-sourced. The degree of ice melt affects both the actual freshwater content and the stratification. Hence, the latter affects the potential for the sea-ice released methane to be retained in the meltwater layer. The sea to air flux is inhibited by the formation of the meltwater layer and increasingly so during its seasonal development (i.e. freshening and warming) and its deepening through wind-induced mixing. The methane excess trapped within this layer is subject to this freshwater discharge and diluted, in a later stage of melt. That is to say, the ongoing ice melt triggers the methane excess formed in PSW, both temporally and spatially. Further work is required to investigate the relevance of more ice-free waters in summer to methane pathways during the melt

season".

- L390-393: You mention warmer waters and Atlantification, Atlantification also changes the vertical ocean stratification in the region. If stratification was to increase, then methane released in surface waters could be trapped close to the surface during summer, leading to potentially increased exchanges with the atmosphere (and transfers into the atmosphere). If stratification was to decrease, methane could spread deeper into the ocean.

It will be changed to:

"For potential long-term consequences, we consider the effects of an increased ocean heat content leading to enhanced ice melt and, hence, more freshwater discharged into the surface layer in the region of the Yermak Plateau. A fresher (and perhaps thicker) surface layer 'cap' than today would cause stronger stratification, inhibiting the exchange between the atmosphere and the subsurface ocean layers. Thus, any methane excess in the waters below this 'cap' would be disconnected from the atmosphere and remains preserved. Within the surface layer itself, however, a larger amount of freshwater could potentially lead to an increased dilution effect. Although the Yermak Plateau area is, in general, identified as a region of large tidal variability and enhanced mixing rates (Fer et al., 2015; Meyer et al., 2017b), these tidal mixing mechanisms are mainly manifested at deeper levels due to the interference between the tidal flow and the highly variable bottom topography. Since our study focuses on the upper 100 m, the effect of tidal mixing is minor. Under the scenario of warmer AW inflow waters, the sink capacity of the surface waters for sea ice released methane may be enhanced, either by dilution in the PSW or by mixing into the deeper ocean. Atlantification may finally contribute to disconnect the meltwater layer and a potential methane excess therein from the atmosphere. Furthermore, enhanced deep winter convection may lead to a weakened stratification, and downwards transport of methane formerly preserved in meltwater layers into the entire PSW. Especially vulnerable for such changes are the areas beyond the current inflow area in the Eurasian basin, where the effect of the
"Atlantification" is expected to be enhanced (Polyakov et al., 2017)".

- L394-398: The overall transfer of methane from sea ice to the ocean stays the same, whether the ice and ocean 'travel' together or not. But changes in how far from the source, the methane is released into the ocean and atmosphere. This you don't mention here.

It will be changed to:

"Finally, the relative velocities of the ice and water and the impacts of stratification on methane signal retention in the surface waters are important factors for the detection of sea ice-induced methane excess in seawater underneath the ice. Tracing the overall transfer of methane from sea ice into the ocean is important for understanding and quantifying the dynamic contribution of sea ice for the methane source-sink balance. It is not yet clear which process contributes the largest amount of methane release from sea ice: the brine release during freeze-up in winter or during melting in spring. Both processes need to be considered and the amount of methane needs to be quantified. Extended analyses and robust numerical modelling of these processes within the entire sea ice-ocean (and atmosphere) system are needed to improve our ability to predict the consequences of the methane source-sink balance modifications in the Arctic Ocean".

-There is also the acceleration of sea ice drift in the Arctic which means that sea ice rich in methane that formed on the Siberian shelves, is now potentially drifting further out with the TPD towards Fram Strait before melting and changing and therefore before releasing its methane.

The following paragraph will be included:

"The type and structure of Arctic sea ice affects the capacity for methane storage (Fig. 9). Our study provides evidence that ridged/rafted sea ice structures create environments where methane oxidation occurs during the Transpolar Drift Stream (TDS), eventually acting as a sink for methane. A faster sea ice drift (Spreen et al., 2011) resulting

from a thinning ice cover may reduce the time for methane to be oxidized within the ice, leading to changes in the methane pathways. Further research should consider rate measurements of methane oxidation mainly in ridged/rafted ice structures to determine the impact of this process in the long-term. On the other hand, with an accelerated sea ice transport, methane up-taken in sea ice will be transported to remote areas, and released in surface waters of regions not yet affected by methane excess. We suggest that future studies should be focused on sea ice formation on different Arctic shelves to validate the importance of methane uptake during ice formation".

-Figure 1: Make it clearer which part of the drift is in Region 1 and which part in Region2.

It will be changed to: "drifting days of region 1 in red and of region 2 in black color."

-Figure 9: Add 'Within ridges / rafted sea ice' for FYI and 'Under ridged / rafted sea ice' for PSW on the figure. The arrow for the atmospheric background signature is not great. Could you not instead have a dot, or create another color rectangle to represent standard local atmospheric ranges?

"The arrow will be changed for a rectangle and within ridges / rafted sea ice' for FYI and 'Under ridged / rafted sea ice' for PSW on the figure will be included".

Once again, thank you very much for the careful revision of the manuscript and valuable feedback. Sincerely on behalf of all co-author. Josefa Verdugo

---

## Author Response (AR1)

*Dear Reviewer 1,*

*Thank you very much for your feedback. We greatly appreciate your helpful comments and suggestions to improve our manuscript.*

*The modifications in regards to your comments made, are shown here accordingly to line numbers of the clean version of the revised manuscript.*

***With respect to your major comments made***

- Results: Most of the results are direct descriptions of the figures without any further take or analysis. It is very repetitive and sometimes hard to link to the further steps you take in the Discussions. It might be useful to move some of the discussion results in the Result section.

*The ice conditions of our floe were very heterogenous, e.g. different snow thickness, ice thickness, ice types/structures, ice permeability, hence we are convinced that a detailed description of all ice cores will be very helpful. Especially for readers interested in comparison of ice cores from later studies with the ice cores presented in our study. To improve the overview of the ice floe characteristics, we included the following text in section 3.1.*

*In L164-166: "Ridged and rafted ice can be especially relevant for the methane cycling, due to the fact that they remain more consolidated even in the summer season (in comparison to FYI) and thus allow us to investigate methane-related processes in certain layers of these ice structures".*

*L125-126: "To highlight the spatial variability of the sea ice physical and biogeochemical properties across the ice floe, we describe each ice core in detail below".*

- Discussion and conclusions: what are your main findings or take-home message? At the moment it is not obvious. For example, in the abstract the last sentence states 'We point to sea ice as a potential source of methane'. This sounds very speculative with 'we point' rather than 'we find'; and with 'a potential'. Either be more assertive about what you have found, or if you must stay speculative, then suggest more alternative theories that explain your observations.

*In the abstract the following sentence: "We point to sea ice as a potential source of methane"*

*In L15-16 it was changed to: "We suggest that sea ice loaded with methane acts as a source of methane for Polar Surface waters during early spring.".*

*In addition to your comment, we included the following take-home messages in the outlook/conclusion section:*

- *In L383-385: "Our study provides evidence that ridged/rafted sea ice structures create environments where methane oxidation occurs during the Transpolar Drift (TPD), eventually acting as a sink for methane".*

- *In L391-392:" For the season of early spring we propose methane release from sea ice into the meltwater layer as predominant pathway. At this time, basal melt is occurring and the top of sea ice loaded with methane is still impermeable".*

- *In L401: "Our study suggests that the excess of methane in PSW during early spring is sea ice sourced".*

- *In L406-410: "The relative velocities of the ice and water, the influence of stratification on methane signal retention in the surface waters, and the impact of mechanical mixing from e.g. winds and tides are important factors for the evolution of sea ice-induced methane excess in seawater underneath the ice".*

-For example, you never mention the potential of the local sediments as sources of methane in the water column, for example nearby or on the Yermak Plateau? Why could this not be the case? Please try to better highlight your key findings throughout the manuscript (abstract, discussion and conclusions).

*In section 4.2 Dissolved methane in Polar surface water (PSW) we included an explanation why we ruled out the sediment sources for the methane excess in PSW, as followed:*

*In L331-335:" In general, methane excess in seawater could also originate from sediments. In our case, a potential source could have been the area West of Svalbard (Sahling et al., 2014; Smith et al., 2014; Westbrook et al., 2009). However, methane released from sediments are laterally transported in the deep ocean and do not reach the surface waters (Damm et al., 2005; Graves et al., 2015; Silyakova et al., 2020). Hence, the PSW remains unaffected by methane released from sediment sources further south. Based on our data and the regional oceanographic conditions, we suggest that methane release from sea ice is a source of the observed excess in PSW".*

**With respect to your minor comments made**

- Some English grammatical and formulation issues, as well as missing words and typos.
Please read and check carefully before resubmission.

*The English grammar and formulation have been checked in the revised version of the manuscript.*

***With respect to your individual comments made***

1.Introduction:

- L25-27: Add somewhere that you are talking about enhanced methane emissions from the ocean to the atmosphere.

*In L24-25 it was changed to: "In particular, sea ice retreat may quickly induce enhanced methane (CH4) emissions from the surface ocean into the atmosphere due to the loss of its barrier function for sea-air gas exchange (Wahlstrom and Meier, 2014)".*

- L27: 'Because the Arctic holds large natural sources of this highly potent...' again ,do you mean the 'Arctic Ocean'?? or sea floor or sediments?

"Because the Arctic holds large natural sources of this highly potent"*: This sentence has been removed from the introduction, we have included instead a paragraph with respect to the methane reservoir in the Siberian shelf waters:*

*In L29-31: "Accordingly, the methane reservoir estimate in the East Siberian and Laptev Seas, ranges from 1.6 and 5.7 Gg CH4 in the seawater, varying with season and depending on the ice cover (Shakhova et al., 2005; McGuire et al., 2009). Hence, in these shallow shelf seas, methane released from the sediment may be entrapped in sea ice during ice formation (Damm et al., 2015)".*

- L37: 'sea ice charged with methane' Consider using 'sea ice loaded with' or another term?

*It was changed to: "sea ice loaded with methane" in the revised manuscript*

- L39: 'during the last years' please add if you can a time period here to better indicate what you mean by 'the last years'.

*In L35-36 it was changed to: "The structure of sea ice transported by the TPD has undergone substantial changes since the early 1980s, shifting from thicker multi-year ice (MYI) to thinner and more fragile first-year ice (FYI; Zamani et al., 2019; Hansen et al., 2013; Maslanik et al., 2011, 2007)".*

3.Result:

- Table 1: For clarity, could you add '-' when you don't have values for the isotopic composition? I expect that for station C9, the 3 values of isotopic composition is 3 different estimates? Can you add that information in the Table caption?

*We have added '-' when there are no values for the isotopic composition.*

*The following sentence was added to the table caption: "Brine samples were taken at stations C8, C10, C11 (one sample per station), and at C9 (three samples").*

- L122: 'has an age of 1 to 3 years, respectively' What do you mean by respectively there?

*In L121-122 it was changed to: "Backward drift trajectories suggest that our floe originated in the Siberian Sea, while the sea ice was estimated to be 1-3 years old (Wollenburg et al., 2020)".*

- Line 196: 'at 90-100 m depth (Fig. 6a and 7a).' You refer to Fig 6 and 7 after Fig2, without having mentioned Fig 3-5. You will have to reorder the figures to match the order you refer to them.

*Figures have been re-ordered in the revised manuscript*

- Figure 7: There are 6 subplots but you only label 5 of them (a-e). The odd one out is the 3rd from the top, which I think is the ice melt estimates? Please add a subplot letter and clarify the figure caption.

*The subplot letter was added, and the following information was included in the caption:*

*"Figure 4: Vertical distribution of several parameters in the upper 100 meters, during the entire drift 4-15 June, 2017; (a) the bars on the panel show the degree of ice melt at each station estimated from the T/S profiles following Peralta-Ferriz and Woodgate (2015). Due to the early melt season, the length scale has been omitted to emphasize that this is only used as a qualitatively guidance".*

4. Discussion:

- Fig.3: Very nice figure. We can guess most of the media but they should still be annotated: Atmosphere, ocean, sea ice, snow? In (IIa), what's the white section with blue dots? Why the gradual change in color of the blue ocean? And do all the CH4 annotation in the ocean indicate concentrations? If so, make it more obvious. Also adda definition of the black arrows in the caption.

*"Atmosphere, ocean, sea ice, snow", were added in the cartoon.*

*The figure caption was changed to:*

*Figure 5: Potential pathways of methane in sea ice with varying impermeable (indicated in grey) and permeable sections (in white with blue dots), i.e. winter (I) and spring (II) conditions. I (a) Relicts of the initial methane signal (source) entrapped in impermeable ice. Impermeable*

*intermediate sea ice layers, act as a barrier for the upward/downward transport of methane (black arrow overlaid by a blue cross). (b) Residual methane signal after methane oxidation occurred in permeable sea ice ("water pocket"), enclosed by impermeable ice layers (see Fig. 6). II (a) When basal melt starts but the top layer still is impermeable and with snow cover (white layer on top of the ice), downward brine transport initiates release of dissolved methane. Flushing events trigger methane released into the ocean. (see chapter 4.1.3). (b) Un-restricted migration of methane in permeable sea ice (black dotted arrow). Ongoing sea ice melt, when freshwater from melted sea ice is released into the water underneath, resulting on a meltwater layer, where methane remains sustained during early spring. Methane (CH₄) annotation indicates concentration. Color gradient in the ocean, reflects the increasing stratification during the seasonal evolution of the upper part of the WML (in blue) into a fresh meltwater layer (in white).*

- L278-279: 'With changes in sea ice dynamics, more of this complex ice structures may be formed, which in turn may promote changes on the methane cycling within sea ice.': here you mean to discuss implications for the future Arctic but its not obvious. Please rephrase.

*In L282-284 it was changed to: "As potential response to the expected future thinning of the sea ice, an increased number of permeable pockets formed during ice ridging may lead to favored methane oxidation therein. Under these circumstances, we suggest that the methane pathways can be modified, i.e., sea ice may be considered as a sink for methane".*

- L343-344: 'In summary, the excess of methane in the surface water clearly point to sea-ice-sourced and early melt events as most important factors for methane release.'. This seems like too strong a statement considering the evidence you have presented.

*In L352-253 it was changed to: "Hence, the methane super-saturation levels in PSW, at this time of the year, is likely to be sea ice-sourced and the ongoing ice melt process influences this excess".*

- Section 4.2.2: some attention needs to be given to the English in this section, with many language choices that are not English based. Eg: 'the more joint journey is made...', 'According the drift direction, one would'

*The English language has been checked in the revised manuscript*

5. Outlook/conclusion

*A significant part of the outlook/conclusion section has been re-written and we included additional information in the revised manuscript.*

- L379: 'We suggest that sea ice methane-released into the ocean, and in this case into the PSW, is the favored pathway in early spring.' Do you mean anywhere? In the whole Arctic? In this region only? Please add details.'

*In L390-397 it was changed to: "For the season of early spring we propose methane release from sea ice into the meltwater layer as predominant pathway. At this time, basal melt is occurring and the top of sea ice loaded with methane is still impermeable. Tracing the overall transfer of methane from sea ice into the ocean is important for understanding and quantifying the dynamic contribution of sea ice for the methane source-sink balance. It is not yet clear which process contributes the largest amount of methane release from sea ice: the brine release during freeze-up in winter or during melting in spring. Both processes need to be considered and the amount of methane must be quantified. Extended analyses and robust numerical modelling of these processes within the entire sea ice-ocean (and atmosphere) system are needed to improve our ability to predict the consequences of the methane source-sink balance modifications in the Arctic Ocean".*

- L385: 'The final fate of the methane (excess) thereafter depends on to which extent it is diluted by additional meltwater.

*In L398-403 it was changed to: "Our study suggests that the excess of methane in PSW during early spring is sea ice sourced. The degree of ice melt regulates this excess through the amount of meltwater added to the surface layer by (a) ruling dilution throughout the melting period (b) affecting the stratification and the potential for the sea-ice released methane to be retained in the meltwater layer. The meltwater layer also inhibits the sea-to-air flux from deeper levels and increasingly so during its seasonal development (i.e. freshening and warming) when it deepens through various mixing processes. Further studies should estimate the amount of methane released into the atmosphere by the sea ice-to-air flux compared to the amount released by brine rejection into the marine environment".*

-What about the dilution by ocean mixing, currents, tides etc..? You don't mention the role of stratification here. - L394-398: The overall transfer of methane from sea ice to the ocean stays the same, whether the ice and ocean 'travel' together or not.

*In L404-406 it was re-phrased to: The relative velocities of the ice and water, the influence of stratification on methane signal retention in the surface waters, and the impact of mechanical mixing from e.g. winds and tides are important factors for the evolution of sea ice-induced methane excess in seawater underneath the ice. Dedicated studies for these processes are needed to better understand their relative importance for this context*

- L390-393: You mention warmer waters and Atlantification, Atlantification also changes the vertical ocean stratification in the region. If stratification was to increase, then methane released in surface waters could be trapped close to the surface during summer, leading to potentially increased exchanges with the atmosphere (and transfers into the atmosphere). If stratification was to decrease, methane could spread deeper into the ocean.

*In L408-417 it was changed to: "Finally, as long-term consequences, we consider the effects of an increased ocean heat content leading to enhanced ice melt and, hence, more freshwater discharged into the surface layer. Within the surface layer itself, a larger amount of freshwater would lead to an increased dilution effect on the methane content. The sink capacity of the surface waters for sea ice released methane may be increased, either by dilution or by mechanical mixing processes. A fresher (and perhaps thicker) surface layer 'cap' than today could further inhibit the exchange of methane between the atmosphere and the subsurface ocean layers through stronger stratification/isolation relative to below waters. Thus, any methane excess in the waters below this 'cap' would be disconnected from the atmosphere and be subject to further mixing with surrounding waters. Especially vulnerable for such changes are the areas beyond the current inflow area in the Eurasian basin, where the effect of the "Atlantification" is expected to be enhanced (Polyakov et al., 2017). Further work is required to investigate the spatial and temporal effects of the expected increase of ice-free waters in summer to methane pathways during the melt season".*

- But changes in how far from the source, the methane is released into the ocean and atmosphere. This you don't mention here. -There is also the acceleration of sea ice drift in the Arctic which means that sea ice rich in methane that formed on the Siberian shelves, is now potentially drifting further out with the TPD towards Fram Strait before melting and changing and therefore before releasing its methane.

*In L384-388 it was changed to: "A faster sea ice drift (Spreen et al., 2011) resulting from a thinning ice cover may reduce the time for methane to be oxidized within the ice, leading to changes in the methane pathways. Further research should consider rate measurements of*

*methane oxidation mainly in ridged/rafted ice structures to determine the long-term impact of this process. On the other hand, with an accelerated sea ice transport, methane taken up in sea ice will be transported to remote areas, and released in surface waters of regions not yet affected by methane excess".*

Figure 1: Make it clearer which part of the drift is in Region 1 and which part in Region2.

*It was changed to: "In red, the stations/dates located over the Yermak Plateau (Region 1) and in black, over the Yermak Plateau eastern flanks (Region 2)".*

Figure 9: Add 'Within ridges / rafted sea ice' for FYI and 'Under ridged / rafted sea ice' for PSW on the figure. The arrow for the atmospheric background signature is not great. Could you not instead have a dot, or create another color rectangle to represent standard local atmospheric ranges?

*The arrow was changed for a rectangle, and your suggestions have been included in the figure*

*Once again many thanks for the careful revision of the manuscript and valuable feedback.*
*With best regards on behalf of all co-author.*
*Josefa Verdugo*

---

## Author Response (AR2)

*Dear Christian Haas,*

*Thank you very much for considering our manuscript for possible publication.*

*We have revised the manuscript with particular attention to the results section and we have modified it in a way to make it more clear.*

**With respect to your minor comments made**

-Structure of the results section can still be improved.

*R: we have worked on this section, in particular in the section 3.1. We carefully modified it to make this result section clearer, as followed:*

**3.1 Sea ice core characteristics**

[revised manuscript text omitted]

Many thanks for the careful revision of the manuscript and valuable feedback.
With best regards on behalf of all co-author.
Josefa Verdugo

---

## Author Response (AR3)

*Dear Christian Haas,*

*Thank you very much for accepting our manuscript for final publication in TC.*

*We have uploaded all required files.*

*With best regards on behalf of all co-author.*
*Josefa Verdugo*